# On the Implicit Bias of Linear Equivariant Steerable Networks

**Ziyu Chen**
Department of Mathematics and Statistics
University of Massachusetts Amherst
Amherst, MA 01003
`ziyuchen@umass.edu`

**Wei Zhu**
Department of Mathematics and Statistics
University of Massachusetts Amherst
Amherst, MA 01003
`weizhu@umass.edu`

## Abstract

We study the implicit bias of gradient flow on linear equivariant steerable networks in group-invariant binary classification. Our findings reveal that the parameterized predictor converges in direction to the unique group-invariant classifier with a maximum margin defined by the input group action. Under a unitary assumption on the input representation, we establish the equivalence between steerable networks and data augmentation. Furthermore, we demonstrate the improved margin and generalization bound of steerable networks over their non-invariant counterparts.

## 1   Introduction

Despite recent theoretical breakthroughs in deep learning, it is still largely unknown why overparameterized deep neural networks (DNNs) with infinitely many solutions achieving near-zero training error can effectively generalize on new data. However, the consistently impressive results of DNNs trained with first-order optimization methods, *e.g.*, gradient descent (GD), suggest that the training algorithm is *implicitly guiding* the model towards a solution with strong generalization performance.

Indeed, recent studies have shown that gradient-based training methods effectively regularize the solution by *implicitly minimizing* a certain complexity measure of the model [Vardi, 2022]. For example, Gunasekar et al. [2018b] showed that in separable binary classification, the linear predictor parameterized by a linear fully-connected network trained under GD converges in the direction of a max-margin support vector machine (SVM), while linear convolutional networks are implicitly regularized by a depth-related bridge penalty in the Fourier domain. Yun et al. [2021] extended this finding to linear tensor networks. Lyu and Li [2020] and Ji and Telgarsky [2020] established the implicit max-margin regularization for (nonlinear) homogeneous DNNs in the parameter space.

On the other hand, another line of research aims to *explicitly regularize* DNNs through architectural design to exploit the inherent structure of the learning problem. In recent years, there has been a growing interest in leveraging group symmetry for this purpose, given its prevalence in both scientific and engineering domains. A significant body of literature has been devoted to designing group-equivariant DNNs that ensure outputs transform covariantly to input symmetry transformations. *Group-equivariant steerable networks* represent a general class of symmetry-preserving models that achieve equivariance with respect to any pair of input-output group actions [Cohen et al., 2019, Weiler and Cesa, 2019, Cohen and Welling, 2017]. Empirical evidence suggests that equivariant steerable networks yield substantial improvements in generalization performance for learning tasks with group symmetry, especially when working with limited amounts of data.

There have been several recent attempts to account for the empirical success of equivariant steerable networks through establishing a tighter upper bound on the test risk for these models. This is typically accomplished by evaluating the complexity measures of equivariant and non-equivariant models under the same norm constraint on the network parameters [Sokolic et al., 2017, Sannai

et al., 2021, Elesedy, 2022]. Nevertheless, it remains unclear whether or why a GD-trained steerable network can achieve a minimizer with a parameter norm comparable to that of its non-equivariant counterpart. Consequently, the effectiveness of such complexity-measure-based arguments to explain the generalization enhancement of steerable networks in group symmetric learning tasks may not be directly applicable.

In light of the above issues, in this work, we aim to fully characterize the implicit bias of the training algorithm on linear equivariant steerable networks in group-invariant binary classification. Our result shows that when trained under gradient flow (GF), *i.e.*, GD with an infinitesimal step size, the steerable-network-parameterized predictor converges in direction to the unique group-invariant classifier attaining a maximum margin with respect to a norm defined by the input group representation. This result has three important implications: under a unitary input group action,

- a linear steerable network trained on the *original* data set converge in the same direction as a linear fully-connected network trained on the *group-augmented* data set. This suggests the equivalence between training with linear steerable networks and *data augmentation*;

- when trained on the same *original* data set, a linear steerable network always attains a wider margin on the *group-augmented* data set compared to a fully-connected network;

- when the underlying distribution is group-invariant, a GF-trained linear steerable network achieves a tighter generalization bound compared to its non-equivariant counterpart. This improvement in generalization is not necessarily dependent on the group size, but rather it depends on the support of the invariant distribution.

Before we end this section, we note that a similar topic has recently been explored by Lawrence et al. [2021] in the context of linear Group Convolutional Neural Networks (G-CNNs), a special case of the equivariant steerable networks considered in this work. However, we point out that the models they studied were not truly group-invariant, and thus their implicit bias result does not explain the improved generalization of G-CNNs. We will further elaborate on the comparison between our work and [Lawrence et al., 2021] in Section 2.

## 2 Related work

**Implicit biases**: Recent studies have shown that for linear regression with the logistic or exponential loss on linearly separable data, the linear predictor under GD/SGD converges in direction to the max-$L^2$-margin SVM [Soudry et al., 2018, Nacson et al., 2019, Gunasekar et al., 2018a]. These results are extended to linear fully-connected networks and linear Convolutional Neural Networks (CNNs) by Gunasekar et al. [2018b] under the assumption of directional convergence and alignment of the network parameters, which are later proved by Ji and Telgarsky [2019a,b], Lyu and Li [2020], Ji and Telgarsky [2020]. The implicit regularization of gradient flow (GF) is further generalized to linear tensor networks by Yun et al. [2021]. For overparameterized nonlinear networks in the infinite-width regime, rigorous analysis on the optimization of DNNs has also been studied from the neural tangent kernel [Jacot et al., 2018, Du et al., 2019, Allen-Zhu et al., 2019] and the mean-field perspectives [Mei et al., 2019, Chizat and Bach, 2018]. However, these models are not explicitly designed to be group-invariant/equivariant, and the implicit bias of gradient-based methods might guide them to converge to sub-optimal solutions in learning tasks with intrinsic group symmetry.

**Equivariant neural networks.** Since their introduction by Cohen and Welling [2016], group equivariant network design has become a burgeoning field with numerous applications from computer vision to scientific computing. Unlike the implicit bias induced by training algorithms, equivariant networks *explicitly* incorporate symmetry priors into model design through either group convolutions [Cheng et al., 2019, Weiler et al., 2018, Sosnovik et al., 2020, Zhu et al., 2022] or, more generally, steerable convolutions [Weiler and Cesa, 2019, Cohen et al., 2019, Worrall et al., 2017]. Despite their empirical success, it remains largely unknown why and whether equivariant networks trained under gradient-based methods actually converge to solutions with smaller test risk in group-symmetric learning tasks [Sokolic et al., 2017, Sannai et al., 2021]. In a recent study, Elesedy and Zaidi [2021] demonstrated a provably strict generalisation benefit for equivariant networks in linear regression. However, the network studied therein is non-equivariant and only symmetrized after training; such test-time data augmentation is different from practical usage of equivariant networks.

**Comparison to Lawrence et al. [2021].** A recent study by Lawrence et al. [2021] also analyzes the implicit bias of linear equivariant G-CNNs, which are a special case of the steerable networks considered in this work. However, the networks studied therein are not truly equivariant/invariant. More specifically, the input space $\mathcal{X}_0$ they considered is the set of functions on the group $G$, *i.e.*, $\mathcal{X}_0 = \{f : G \to \mathbb{R}\}$, and the input $G$-action is given by the regular representation

$$[\rho_0(g)f](g') = f(g^{-1}g'), \quad \forall g \in G, \forall f \in \mathcal{X}_0. \tag{1}$$

In their case, due to the transitivity of the regular representation (1), $G$-invariant linear functions $\Phi : \mathcal{X}_0 \to \mathbb{R}$ are constrained to the (trivial) form:

$$\Phi(f) = C \sum_{g \in G} f(g),$$

where $C$ is a multiplicative constant. To avoid learning this trivial function, Lawrence et al. [2021] chose to parameterize $\Phi$ using a linear G-CNN, in which the final layer is replaced by a fully-connected layer. While this fully-connected layer provides the capability to learn more complex and nontrivial functions, it simultaneously undermines the property of group invariance. Therefore, their implicit bias result does not explain the improved generalization of G-CNNs. In contrast, we assume Euclidean inputs with non-transitive group actions, allowing linear steerable networks to learn non-trivial group-invariant models.

## 3 Background and problem setup

We provide a brief background in group theory and group-equivariant steerable networks. We also explain the setup of our learning problem for group-invariant binary classification.

### 3.1 Group and group equivariance

A *group* is a set $G$ equipped with a binary operator, the group product, satisfying the axioms of associativity, identity, and invertibility. We always assume in this work that $G$ is finite, *i.e.*, $|G| < \infty$, where $|G|$ denotes its cardinality.

Given a vector space $\mathcal{X}$, let $\mathrm{GL}(\mathcal{X})$ be the general linear group of $\mathcal{X}$ consisting of all invertible linear transformations on $\mathcal{X}$. A map $\rho : G \to \mathrm{GL}(\mathcal{X})$ is called a *group representation* (or *linear group action*) of $G$ on $\mathcal{X}$ if $\rho$ is a group homomorphism from $G$ to $\mathrm{GL}(\mathcal{X})$, namely

$$\rho(gh) = \rho(g)\rho(h) \in \mathrm{GL}(\mathcal{X}), \quad \forall g, h \in G. \tag{2}$$

When the representation $\rho$ is clear from the context, we also abbreviate the group action $\rho(g)\mathbf{x}$ as $g\mathbf{x}$.

Given a pair of vector spaces $\mathcal{X}, \mathcal{Y}$ and their respective $G$-representations $\rho_\mathcal{X}$ and $\rho_\mathcal{Y}$, a linear map $\Psi : \mathcal{X} \to \mathcal{Y}$ is said to be *G-equivariant* if it commutes with the $G$-representations $\rho_\mathcal{X}$ and $\rho_\mathcal{Y}$, *i.e.*,

$$\Psi \circ \rho_\mathcal{X}(g) = \rho_\mathcal{Y}(g) \circ \Psi, \quad \forall g \in G. \tag{3}$$

Linear equivariant maps are also called *intertwiners*, and we denote by $\mathrm{Hom}_G(\rho_\mathcal{X}, \rho_\mathcal{Y})$ the space of all intertwiners satisfying (3). When $\rho_\mathcal{Y} \equiv \mathrm{Id}$ is the trivial representation, then $\Psi[\rho_\mathcal{X}(g)(\mathbf{x})] = \Psi[\mathbf{x}]$ for all $\mathbf{x} \in \mathcal{X}$; namely, $\Psi$ becomes a $G$-invariant linear map.

### 3.2 Equivariant steerable networks and G-CNNs

Let $\mathcal{X}_0 = \mathbb{R}^{d_0}$ be a $d_0$-dimensional input Euclidean space, equipped with the usual inner product. Let $\rho_0 : G \to \mathrm{GL}(\mathcal{X}_0)$ be a $G$-representation on $\mathcal{X}_0$. Suppose we have an unknown target function $f^* : \mathcal{X}_0 = \mathbb{R}^{d_0} \to \mathbb{R}$ that is $G$-invariant under $\rho_0$, *i.e.*, $f^*(g\mathbf{x}) := f^*(\rho_0(g)\mathbf{x}) = f^*(\mathbf{x})$ for all $g \in G$ and $\mathbf{x} \in \mathcal{X}_0$. The goal of equivariant steerable networks is to approximate $f^*$ using an $L$-layer neural network $f = \Psi_L \circ \Psi_{L-1} \circ \cdots \circ \Psi_1$ that is guaranteed to be also $G$-invariant.

Since the composition of equivariant maps is also equivariant, it suffices to specify a collection of $G$-representation spaces $\{(\mathcal{X}_l, \rho_l)\}_{l=1}^L$, with $(\mathcal{X}_L, \rho_L) = (\mathbb{R}, \mathrm{Id})$ being the trivial representation, such that each layer $\Psi_l \in \mathrm{Hom}_G(\rho_{l-1}, \rho_L) : \mathcal{X}_{l-1} \to \mathcal{X}_l$ is $G$-equivariant. Equivalently, we want the

following diagram to be commutative,

$$
\begin{array}{ccccccccccc}
\mathcal{X}_0 & \xrightarrow{\Psi_1} & \mathcal{X}_1 & \xrightarrow{\Psi_2} & \mathcal{X}_2 & \xrightarrow{\Psi_3} & \cdots & \xrightarrow{\Psi_{L-1}} & \mathcal{X}_{L-1} & \xrightarrow{\Psi_L} & \mathcal{X}_L \\
\rho_0(g)\Big\downarrow & & \rho_1(g)\Big\downarrow & & \rho_2(g)\Big\downarrow & & & & \rho_{L-1}(g)\Big\downarrow & & \rho_L(g)\Big\downarrow \\
\mathcal{X}_0 & \xrightarrow{\Psi_1} & \mathcal{X}_1 & \xrightarrow{\Psi_2} & \mathcal{X}_2 & \xrightarrow{\Psi_3} & \cdots & \xrightarrow{\Psi_{L-1}} & \mathcal{X}_{L-1} & \xrightarrow{\Psi_L} & \mathcal{X}_L
\end{array}
\quad , \quad \forall g \in G. \tag{4}
$$

**Equivariant steerable networks.** Given the representations $\{(\mathcal{X}_l, \rho_l)\}_{l=0}^{L}$, a (linear) *steerable network* is constructed as follows. For each $l \in [L] := \{1, \cdots, L\}$, we choose a finite collection of $N_l$ (pre-computed) intertwiners $\{\psi_l^j\}_{j=1}^{N_l} \subset \mathrm{Hom}_G(\rho_{l-1}, \rho_l)$. Typically, $\{\psi_l^j\}_{j=1}^{N_l}$ is a basis of $\mathrm{Hom}_G(\rho_{l-1}, \rho_l)$, but it is not necessary in our setting. The $l$-th layer equivariant map $\Psi_l^{\mathrm{steer}} : \mathcal{X}_{l-1} \to \mathcal{X}_l$ of a steerable network is then parameterized by

$$
\Psi_l^{\mathrm{steer}}(\mathbf{x}; \mathbf{w}_l) = \sum_{j \in [N_l]} w_l^j \psi_l^j(\mathbf{x}), \quad \forall \mathbf{x} \in \mathcal{X}_{l-1}, \tag{5}
$$

where the coefficients $\mathbf{w}_l = [w_l^j]_{j \in [N_l]}^\top \in \mathbb{R}^{N_l}$ are the trainable parameters of the $l$-th layer. An $L$-layer linear steerable network $f_{\mathrm{steer}}(\mathbf{x}; \mathbf{W})$ is then defined as the composition

$$
f_{\mathrm{steer}}(\mathbf{x}; \mathbf{W}) = \Psi_L^{\mathrm{steer}}(\cdots \Psi_2^{\mathrm{steer}}(\Psi_1^{\mathrm{steer}}(\mathbf{x}; \mathbf{w}_1); \mathbf{w}_2) \cdots ; \mathbf{w}_L), \tag{6}
$$

where $\mathbf{W} = [\mathbf{w}_l]_{l=1}^{L} \in \prod_{l=1}^{L} \mathbb{R}^{N_l} =: \mathcal{W}_{\mathrm{steer}}$ is the collection of all trainable parameters. The network $f_{\mathrm{steer}}(\mathbf{x}; \mathbf{W})$ defined in (6) is referred to as a *steerable network* because it steers the layer-wise output to transform according to any specified representation.

**G-CNNs.** A special case of the steerable networks is the *group convolutional neural network* (G-CNN), wherein the hidden representation space $(\mathcal{X}_l, \rho_l)$ for each $l \in [L-1]$ is set to

$$
\mathcal{X}_l = (\mathbb{R}^{d_l})^G = \{\mathbf{x}_l : G \to \mathbb{R}^{d_l}\}, \quad \rho_l(g)\mathbf{x}_l(h) := \mathbf{x}_l(g^{-1}h) \in \mathbb{R}^{d_1}, \forall g, h \in G. \tag{7}
$$

The representation $\rho_l \in \mathrm{GL}(\mathcal{X}_l)$ in (7) is known as the *regular representation* of $G$. Intuitively, $\mathbf{x}_l \in \mathcal{X}_l$ can be viewed as a matrix of size $d_l \times |G|$, and $\rho_l(g)$ is a permutation of the columns of $\mathbf{x}_l$.

With this choice of $\{(\mathcal{X}_l, \rho_l)\}_{l=0}^{L}$, the first-layer equivariant map of a G-CNN is defined as

$$
\Psi_1^{\mathrm{G\text{-}CNN}}(\mathbf{x}; \mathbf{w}_1)(g) = \mathbf{w}_1^\top g^{-1}\mathbf{x} \in \mathbb{R}^{d_1}, \quad \forall \mathbf{x} \in \mathbb{R}^{d_0}, \tag{8}
$$

where $\mathbf{w}_1 = (w_1^{j,k})_{j,k} \in \mathbb{R}^{d_0 \times d_1}$ are the trainable parameters of the first layer. Eq. (8) is called a *G-lifting map* as it lifts a Euclidean signal $\mathbf{x} \in \mathbb{R}^{d_0}$ to a function $\Psi_1^{\mathrm{G\text{-}CNN}}(\mathbf{x}; \mathbf{w}_1)$ on $G$. For the subsequent layers, equivariance is achieved through *group convolutions*, and the readers are referred to the appendix for a detailed explanation.

**Assumptions on the representations.** The input representation $\rho_0 \in \mathrm{GL}(\mathcal{X}_0 = \mathbb{R}^{d_0})$ is assumed to be given, and $\rho_L \equiv 1 \in \mathrm{GL}(\mathcal{X}_L = \mathbb{R})$ is set to the trivial representation for group-invariant outputs. In this work, we make the following special choices for the first-layer representation $(\rho_1, \mathcal{X}_1)$ as well as the equivariant map $\Psi_1^{\mathrm{steer}}(\mathbf{x}, \mathbf{w}_1)$ in a general steerable network.

**Assumption 3.1.** *We adopt the regular representation* (7) *for the first layer, and set the first-layer equivariant map* $\Psi_1^{\mathrm{steer}}(\mathbf{x}; \mathbf{w}_1)$ *to the G-lifting map* (8).

**Remark 3.2.** *The rationale of Assumption 3.1 is for the network to have enough capacity to parameterize any G-invariant linear classifier; this will be further explained in Proposition 3.6. We note that the representations* $(\rho_l, \mathcal{X}_l)$ *and steerable maps* $\Psi_l^{\mathrm{steer}}(\mathbf{x}, \mathbf{w}_l)$ *for all subsequent layers, $l \in \{2, \cdots, L-1\}$, can be arbitrary.*

Under Assumption 3.1, we can characterize the linear steerable networks $f_{\mathrm{steer}}(\cdot; \mathbf{W})$ in the following proposition.

**Proposition 3.3.** *Let* $f_{\mathrm{steer}}(\mathbf{x}; \mathbf{W})$ *be the linear steerable network satisfying Assumption 3.1, where* $\mathbf{W} = [\mathbf{w}_l]_{l=1}^{L} \in \mathcal{W}_{\mathrm{steer}}$ *is the collection of all model parameters. There exists a multi-linear map* $M : (\mathbf{w}_2, \cdots, \mathbf{w}_L) \mapsto M(\mathbf{w}_2, \cdots, \mathbf{w}_L) \in \mathbb{R}^{d_1}$ *such that for all* $\mathbf{x} \in \mathbb{R}^{d_0}$ *and* $\mathbf{W} \in \mathcal{W}_{\mathrm{steer}}$,

$$
f_{\mathrm{steer}}(\mathbf{x}; \mathbf{W}) = f_{\mathrm{steer}}(\bar{\mathbf{x}}; \mathbf{W}) = \langle \bar{\mathbf{x}}, \mathbf{w}_1 M(\mathbf{w}_2, \cdots, \mathbf{w}_L) \rangle, \tag{9}
$$

*where* $\bar{\mathbf{x}} := \frac{1}{|G|} \sum_{g \in G} g\mathbf{x}$ *is the average of all elements on the group orbit of* $\mathbf{x} \in \mathbb{R}^{d_0}$.

Although a straightforward proof of Proposition 3.3 can be readily derived using Schur's Lemma, we have opted to include an elementary proof of Proposition 3.3 in the appendix for the sake of completeness. If we define

$$\overline{\mathcal{P}}_{\text{steer}}(\mathbf{W}) := \mathbf{w}_1 M(\mathbf{w}_2, \cdots, \mathbf{w}_L), \quad \mathcal{P}_{\text{steer}}(\mathbf{W}) := \left( \frac{1}{|G|} \sum_{g \in G} g^\top \right) \overline{\mathcal{P}}_{\text{steer}}(\mathbf{W}), \quad (10)$$

where $g^\top := \rho_0(g)^\top$, then

$$f_{\text{steer}}(\mathbf{x}; \mathbf{W}) = \langle \mathbf{x}, \mathcal{P}_{\text{steer}}(\mathbf{W}) \rangle = \langle \overline{\mathbf{x}}, \overline{\mathcal{P}}_{\text{steer}}(\mathbf{W}) \rangle. \quad (11)$$

By the multi-linearity of $M(\mathbf{w}_2, \cdots, \mathbf{w}_L)$, one can verify that $f_{\text{steer}}(\mathbf{x}; \mathbf{W})$, $\mathcal{P}_{\text{steer}}(\mathbf{W})$, and $\overline{\mathcal{P}}_{\text{steer}}(\mathbf{W})$ are all *L-homogeneous* in $\mathbf{W}$; that is, for all $\nu > 0$ and $\mathbf{W} \in \mathcal{W}_{\text{steer}}$,

$$f_{\text{steer}}(\mathbf{x}; \nu\mathbf{W}) = \nu^L f_{\text{steer}}(\mathbf{x}; \mathbf{W}), \ \mathcal{P}_{\text{steer}}(\nu\mathbf{W}) = \nu^L \mathcal{P}_{\text{steer}}(\mathbf{W}), \ \overline{\mathcal{P}}_{\text{steer}}(\nu\mathbf{W}) = \nu^L \overline{\mathcal{P}}_{\text{steer}}(\mathbf{W}). \quad (12)$$

**Remark 3.4.** *In comparison, an L-layer linear fully-connected network $f_{\text{fc}}(\mathbf{x}; \mathbf{W})$ is given by*

$$f_{\text{fc}}(\mathbf{x}; \mathbf{W}) = \mathbf{w}_L^\top \mathbf{w}_{L-1}^\top \cdots \mathbf{w}_1^\top \mathbf{x} = \langle \mathbf{x}, \mathcal{P}_{\text{fc}}(\mathbf{W}) \rangle, \quad \mathcal{P}_{\text{fc}}(\mathbf{W}) := \mathbf{w}_1 \cdots \mathbf{w}_L. \quad (13)$$

*where $\mathbf{W} = [\mathbf{w}_l]_{l=1}^L \in \mathcal{W}_{\text{fc}} := \left( \prod_{l=1}^{L-1} \mathbb{R}^{d_{l-1} \times d_l} \right) \times \mathbb{R}^{d_{L-1}}$. It is worth noting that when $G = \{e\}$ is the trivial group, a linear fully-connected network $f_{\text{fc}}(\mathbf{x}; \mathbf{W})$ is identical with a linear G-CNN, which is a special case of linear steerable networks. See Remark A.1 for details.*

### 3.3  Group-invariant binary classification

Consider a binary classification data set $S = \{(\mathbf{x}_i, y_i) : i \in [n]\}$, where $\mathbf{x}_i \in \mathbb{R}^{d_0}$ and $y_i \in \{\pm 1\}, \forall i \in [n]$. We assume that $S$ are i.i.d. samples from a *G-invariant distribution* $\mathcal{D}$ defined below.

**Definition 3.5.** *A distribution $\mathcal{D}$ over $\mathbb{R}^{d_0} \times \{\pm 1\}$ is said to be G-invariant with respect to a representation $\rho_0 \in GL(\mathbb{R}^{d_0})$ if*

$$(\rho_0(g) \otimes \text{Id})_* \mathcal{D} = \mathcal{D}, \ \forall g \in G, \quad (14)$$

*where $(\rho_0(g) \otimes \text{Id})(\mathbf{x}, y) := (\rho_0(g)\mathbf{x}, y)$, and $(\rho_0(g) \otimes \text{Id})_* \mathcal{D} := \mathcal{D} \circ (\rho_0(g) \otimes \text{Id})^{-1}$ is the push-forward measure of $\mathcal{D}$ under $\rho_0(g) \otimes \text{Id}$.*

It is easy to verify that the Bayes optimal classifier $f^* : \mathbb{R}^{d_0} \to \{\pm 1\}$ (the one achieving the smallest population risk) for a $G$-invariant distribution $\mathcal{D}$ is necessarily a $G$-invariant function, *i.e.*, $f^*(g\mathbf{x}) = f^*(\mathbf{x}), \forall g \in G$. Therefore, to learn $f^*$ using (linear) neural networks, it is natural to approximate $f^*$ using an equivariant steerable network

$$f^*(\mathbf{x}) \approx \text{sign}\left( f_{\text{steer}}(\mathbf{x}; \mathbf{W}) \right) = \text{sign}\left( \langle \mathbf{x}, \mathcal{P}_{\text{steer}}(\mathbf{W}) \rangle \right). \quad (15)$$

After choosing the exponential loss $\ell_{\text{exp}} : \mathbb{R} \times \{\pm 1\} \to \mathbb{R}_+$, $\ell_{\text{exp}}(\hat{y}, y) := \exp(-\hat{y}y)$, as a surrogate loss function, the empirical risk minimization over $S$ for the steerable network $f_{\text{steer}}(\mathbf{x}; \mathbf{W})$ becomes

$$\min_{\mathbf{W} \in \mathcal{W}_{\text{steer}}} \mathcal{L}_{\mathcal{P}_{\text{steer}}}(\mathbf{W}; S) := \sum_{i=1}^n \ell_{\text{exp}}(\langle \mathbf{x}_i, \mathcal{P}_{\text{steer}}(\mathbf{W}) \rangle, y_i) = \sum_{i=1}^n \ell_{\text{exp}}(\langle \overline{\mathbf{x}}_i, \overline{\mathcal{P}}_{\text{steer}}(\mathbf{W}) \rangle, y_i). \quad (16)$$

On the other hand, since $\mathcal{P}_{\text{steer}}(\mathbf{W}) = \boldsymbol{\beta} \in \mathbb{R}^{d_0}$ always corresponds to a *G-invariant* linear predictor—we have slightly abused the notation by identifying $\boldsymbol{\beta} \in \mathbb{R}^{d_0}$ with the map $\mathbf{x} \mapsto \langle \mathbf{x}, \boldsymbol{\beta} \rangle$—one can alternatively consider the empirical risk minimization directly over the invariant linear predictors $\boldsymbol{\beta}$:

$$\min_{\boldsymbol{\beta} \in \mathbb{R}_G^{d_0}} \mathcal{L}(\boldsymbol{\beta}; S) := \sum_{i=1}^n \ell_{\text{exp}}(\langle \mathbf{x}_i, \boldsymbol{\beta} \rangle, y_i), \quad (17)$$

where $\mathbb{R}_G^{d_0} \subset \mathbb{R}^{d_0}$ is the subspace of all $G$-invariant linear predictors, which is characterized by the following proposition.

**Proposition 3.6.** *Let $\mathbb{R}_G^{d_0} \subset \mathbb{R}^{d_0}$ be the subspace of G-invariant linear predictors, i.e., $\mathbb{R}_G^{d_0} = \{ \boldsymbol{\beta} \in \mathbb{R}^{d_0} : \boldsymbol{\beta}^\top \mathbf{x} = \boldsymbol{\beta}^\top g\mathbf{x}, \forall \mathbf{x} \in \mathbb{R}^{d_0}, \forall g \in G \}$. Then*

*(a)* $\mathbb{R}_G^{d_0}$ *is characterized by*

$$\mathbb{R}_G^{d_0} = \bigcap_{g \in G} \ker(I - g^\top) = \mathrm{Range}\left(\frac{1}{|G|} \sum_{g \in G} g^\top\right). \tag{18}$$

*(b) Let* $\mathcal{A} : \mathbb{R}^{d_0} \to \mathbb{R}^{d_0}$ *be the group-averaging map,*

$$\mathcal{A}(\boldsymbol{\beta}) := \overline{\boldsymbol{\beta}} = \frac{1}{|G|} \sum_{g \in G} g\boldsymbol{\beta}. \tag{19}$$

*Then its adjoint* $\mathcal{A}^\top : \boldsymbol{\beta} \mapsto \frac{1}{|G|} \sum_{g \in G} g^\top \boldsymbol{\beta}$ *is a projection operator from* $\mathbb{R}^{d_0}$ *to* $\mathbb{R}_G^{d_0}$. *In other words,* $\mathrm{Range}(\mathcal{A}^\top) = \mathbb{R}_G^{d_0}$ *and* $\mathcal{A}^\top \circ \mathcal{A}^\top = \mathcal{A}^\top$.

*(c) If* $G$ *acts unitarily on* $\mathcal{X}_0$, *i.e.,* $\rho_0(g^{-1}) = \rho_0(g)^\top$, *then* $\mathcal{A} = \mathcal{A}^\top$ *is self-adjoint. This implies that* $\mathcal{A} : \boldsymbol{\beta} \mapsto \overline{\boldsymbol{\beta}}$ *is an orthogonal projection from* $\mathbb{R}^{d_0}$ *onto* $\mathbb{R}_G^{d_0}$. *In particular, we have*

$$\overline{\boldsymbol{\beta}} = \boldsymbol{\beta} \iff \boldsymbol{\beta} \in \mathbb{R}_G^{d_0}, \quad and \; \|\overline{\boldsymbol{\beta}}\| \leq \|\boldsymbol{\beta}\|, \forall \boldsymbol{\beta} \in \mathbb{R}^{d_0}. \tag{20}$$

Proposition 3.6 combined with (10) demonstrates that a linear steerable network $f_{\mathrm{steer}}(\cdot; \mathbf{W}) = \langle \cdot, \mathcal{P}_{\mathrm{steer}}(\mathbf{W}) \rangle$ can realize any $G$-invariant linear predictor $\boldsymbol{\beta} \in \mathbb{R}_G^{d_0}$; that is, $\{\mathcal{P}_{\mathrm{steer}}(\mathbf{W}) : \mathbf{W} \in \mathcal{W}_{\mathrm{steer}}\} = \mathbb{R}_G^{d_0}$. Therefore (16) and (17) are equivalent optimization problems parameterized in different ways. However, minimizing (16) using gradient-based methods may potentially lead to different classifiers compared to those obtained from optimizing (17) directly.

**Gradient flow.** Given an initialization $\mathbf{W}(0) \in \mathcal{W}_{\mathrm{steer}}$, the gradient flow $\{\mathbf{W}(t)\}_{t \geq 0}$ for (16) is the solution of the following ordinary differential equation (ODE),

$$\frac{d\mathbf{W}}{dt} = -\nabla_{\mathbf{W}} \mathcal{L}_{\mathcal{P}_{\mathrm{steer}}}(\mathbf{W}; S) = -\nabla_{\mathbf{W}} \left[\sum_{i=1}^n \ell_{\exp}(\langle \mathbf{x}_i, \mathcal{P}_{\mathrm{steer}}(\mathbf{W}) \rangle, y_i)\right]. \tag{21}$$

The purpose of this work is to inspect the asymptotic behavior of the $G$-invariant linear predictors $\boldsymbol{\beta}_{\mathrm{steer}}(t) = \mathcal{P}_{\mathrm{steer}}(\mathbf{W}(t))$ parameterized by the linear steerable network trained under gradient flow (21). In particular, we aim to analyze the directional limit of $\boldsymbol{\beta}_{\mathrm{steer}}(t)$ as $t \to \infty$, *i.e.*, $\lim_{t \to \infty} \frac{\boldsymbol{\beta}_{\mathrm{steer}}(t)}{\|\boldsymbol{\beta}_{\mathrm{steer}}(t)\|}$. Before ending this section, we make the following assumption on the gradient flow $\mathbf{W}(t)$ that also appears in the prior works Ji and Telgarsky [2020], Lyu and Li [2020], Yun et al. [2021].

**Assumption 3.7.** *The gradient flow* $\mathbf{W}(t)$ *satisfies* $\mathcal{L}_{\mathcal{P}_{\mathrm{steer}}}(\mathbf{W}(t_0); S) < 1$ *for some* $t_0 > 0$.

This assumption implies that the data set $S = \{(\mathbf{x}_i, y_i) : i \in [n]\}$ can be separated by a $G$-invariant linear predictor $\mathcal{P}_{\mathrm{steer}}(\mathbf{W}(t_0))$, and our analysis is focused on the "late phase" of the gradient flow training as $t \to \infty$.

## 4 Implicit bias of linear steerable networks

Our main result on the implicit bias of gradient flow on linear steerable networks in binary classification is summarized in the following theorem.

**Theorem 4.1.** *Under Assumption 3.1 and Assumption 3.7, let* $\boldsymbol{\beta}_{\mathrm{steer}}(t) = \mathcal{P}_{\mathrm{steer}}(\mathbf{W}(t))$ *be the time-evolution of the* $G$-*invariant linear predictors parameterized by a linear steerable network trained with gradient flow on the data set* $S = \{(\mathbf{x}_i, y_i) : i \in [n]\}$; *cf. Eq. (21). Then*

*(a) The directional limit* $\boldsymbol{\beta}_{\mathrm{steer}}^\infty = \lim_{t \to \infty} \frac{\boldsymbol{\beta}_{\mathrm{steer}}(t)}{\|\boldsymbol{\beta}_{\mathrm{steer}}(t)\|}$ *exists and* $\boldsymbol{\beta}_{\mathrm{steer}}^\infty \propto \frac{1}{|G|} \sum_{g \in G} g^\top \boldsymbol{\gamma}^*$, *where* $\boldsymbol{\gamma}^*$ *is the max-*$L^2$*-margin SVM solution for the **transformed** data* $\overline{S} = \{(\overline{\mathbf{x}}_i, y_i) : i \in [n]\}$:

$$\boldsymbol{\gamma}^* = \arg\min_{\boldsymbol{\gamma} \in \mathbb{R}^{d_0}} \|\boldsymbol{\gamma}\|^2, \quad \text{s.t. } y_i \langle \overline{\mathbf{x}}_i, \boldsymbol{\gamma} \rangle \geq 1, \forall i \in [n]. \tag{22}$$

*Furthermore, if* $G$ *acts unitarily on the input space* $\mathcal{X}_0$, *i.e.,* $g^{-1} = g^\top$, *then* $\boldsymbol{\beta}_{\mathrm{steer}}^\infty \propto \boldsymbol{\gamma}^*$.

*(b) Equivalently, $\boldsymbol{\beta}_{\text{steer}}^{\infty}$ is proportional to the unique minimizer $\boldsymbol{\beta}^*$ of the problem*

$$\boldsymbol{\beta}^* = \arg\min_{\boldsymbol{\beta} \in \mathbb{R}^{d_0}} \|\text{Proj}_{\text{Range}(\mathcal{A})}\boldsymbol{\beta}\|^2, \quad \text{s.t. } \boldsymbol{\beta} \in \mathbb{R}_G^{d_0}, \text{ and } y_i \langle \mathbf{x}_i, \boldsymbol{\beta} \rangle \geq 1, \forall i \in [n], \quad (23)$$

*where $\text{Proj}_{\text{Range}(\mathcal{A})}$ is the projection from $\mathbb{R}^{d_0}$ to $\text{Range}(\mathcal{A}) = \text{Range}\left(\frac{1}{|G|}\sum_{g \in G} g\right)$. More-over, if $G$ acts unitarily on $\mathcal{X}_0$, then*

$$\boldsymbol{\beta}_{\text{steer}}^{\infty} \propto \boldsymbol{\beta}^* = \arg\min_{\boldsymbol{\beta} \in \mathbb{R}^{d_0}} \|\boldsymbol{\beta}\|^2, \quad \text{s.t. } \boldsymbol{\beta} \in \mathbb{R}_G^{d_0}, \text{ and } y_i \langle \mathbf{x}_i, \boldsymbol{\beta} \rangle \geq 1, \forall i \in [n]. \quad (24)$$

*Namely, $\boldsymbol{\beta}_{\text{steer}}^{\infty}$ achieves the maximum $L^2$-margin among all $G$-invariant linear predictors.*

Theorem 4.1 suggests that gradient flow implicitly guides a linear steerable network toward the unique $G$-invariant classifier with a maximum margin defined by the input representation $\rho_0$.

**Remark 4.2.** *According to Remark 3.4, when $G = \{e\}$ is a trivial group, then a linear G-CNN (which is a special case of linear steerable networks) reduces to a fully-connected network $f_{\text{fc}}(\mathbf{x}; \mathbf{W})$. Since the representation of a trivial group is always unitary, we have the following corollary which also appeared in Ji and Telgarsky [2020] and Yun et al. [2021].*

**Corollary 4.3.** *Let $\{\mathbf{W}(t)\}_{t \geq 0} \subset \mathcal{W}_{\text{fc}}$ be the gradient flow of the parameters when training a linear fully-connected network on the data set $S = \{(\mathbf{x}_i, y_i), i \in [n]\}$, i.e.,*

$$\frac{d\mathbf{W}}{dt} = -\nabla_{\mathbf{W}}\mathcal{L}_{\mathcal{P}_{\text{fc}}}(\mathbf{W}; S) := -\nabla_{\mathbf{W}}\left[\sum_{i=1}^{n} \ell_{\exp}(\langle \mathbf{x}_i, \mathcal{P}_{\text{fc}}(\mathbf{W})\rangle, y_i)\right]. \quad (25)$$

*Then the classifier $\boldsymbol{\beta}_{\text{fc}}(t) = \mathcal{P}_{\text{fc}}(\mathbf{W}(t))$ converges in a direction that aligns with the max-$L^2$-margin SVM solution $\boldsymbol{\gamma}^*$ for the **original** data set $S = \{(\mathbf{x}_i, y_i) : i \in [n]\}$,*

$$\boldsymbol{\gamma}^* = \arg\min_{\boldsymbol{\gamma} \in \mathbb{R}^{d_0}} \|\boldsymbol{\gamma}\|^2, \quad \text{s.t. } y_i \langle \mathbf{x}_i, \boldsymbol{\gamma} \rangle \geq 1, \forall i \in [n]. \quad (26)$$

**Remark 4.4.** *While Theorem 4.1 provides a complete characterization of the implicit bias exhibited by gradient flow in linear steerable networks, it is imperative to note that the convergence rate to the directional limit is, in fact, exponentially slow. This is consistent with the findings in, e.g., [Soudry et al., 2018, Yun et al., 2021]. A comprehensive analysis of gradient flow behavior in a non-asymptotic regime falls outside the scope of this current study.*

## 5 The equivalence between steerable networks and data augmentation

Compared to hard-wiring symmetry priors into model architectures through equivariant steerable networks, an alternative approach to incorporate symmetry into the learning process is by training a non-equivariant model with the aid of *data augmentation*. In this section, we demonstrate that these two approaches are equivalent for binary classification under a unitary assumption for $\rho_0$.

**Corollary 5.1.** *Let $\boldsymbol{\beta}_{\text{steer}}^{\infty} = \lim_{t \to \infty} \frac{\boldsymbol{\beta}_{\text{steer}}(t)}{\|\boldsymbol{\beta}_{\text{steer}}(t)\|}$ be the directional limit of the linear predictor $\boldsymbol{\beta}_{\text{steer}}(t) = \mathcal{P}_{\text{steer}}(\mathbf{W}(t))$ parameterized by a linear steerable network trained using gradient flow on the **original** data set $S = \{(\mathbf{x}_i, y_i), i \in [n]\}$. Correspondingly, let $\boldsymbol{\beta}_{\text{fc}}^{\infty} = \lim_{t \to \infty} \frac{\boldsymbol{\beta}_{\text{fc}}(t)}{\|\boldsymbol{\beta}_{\text{fc}}(t)\|}$, $\boldsymbol{\beta}_{\text{fc}}(t) = \mathcal{P}_{\text{fc}}(\mathbf{W}(t))$ (13), be that of a linear fully-connected network trained on the **augmented** data set $S_{\text{aug}} = \{(g\mathbf{x}_i, y_i), i \in [n], g \in G\}$. If $G$ acts unitarily on $\mathcal{X}_0$, then*

$$\boldsymbol{\beta}_{\text{steer}}^{\infty} = \boldsymbol{\beta}_{\text{fc}}^{\infty}. \quad (27)$$

*In other words, the effect of using a linear steerable network for group-invariant binary classification is exactly the same as conducting **data-augmentation** for non-invariant models.*

**Remark 5.2.** *The equivalence between data augmentation and training with a linear steerable network is valid only in an asymptotic sense, yet the underlying training dynamics differ substantially. Specifically, $\boldsymbol{\beta}_{\text{steer}}(t)$ is assured to maintain $G$-invariance throughout the training process, whereas $\boldsymbol{\beta}_{\text{fc}}(t)$ achieves $G$-invariance solely in the limiting case as $t \to \infty$. Moreover, the equivalence only holds for "full-batch" data-augmentation over the entire group orbits.*

**Remark 5.3.** *For Corollary 5.1 to hold, it is crucial that $\rho_0 \in \mathrm{GL}(\mathcal{X}_0)$ is unitary, as otherwise the limit direction $\boldsymbol{\beta}_{\mathrm{fc}}^{\infty}$ of a linear fully-connected network trained on the augmented data set is generally not $G$-invariant (and hence cannot be equal to $\boldsymbol{\beta}_{\mathrm{steer}}^{\infty} \in \mathbb{R}_G^{d_0}$); see Example 5.4.*

**Example 5.4.** Let $G = \mathbb{Z}_2 = \{\overline{0}, \overline{1}\}$. Consider a (non-unitary) $G$-representation $\rho_0$ on $\mathcal{X}_0 = \mathbb{R}^2$,

$$\rho_0\left(\overline{0}\right) = \begin{bmatrix} 1 & 0 \\ 0 & 1 \end{bmatrix}, \quad \rho_0\left(\overline{1}\right) = \begin{bmatrix} 1 & 0 \\ -1 & 1 \end{bmatrix} \begin{bmatrix} -1 & 0 \\ 0 & 1 \end{bmatrix} \begin{bmatrix} 1 & 0 \\ 1 & 1 \end{bmatrix} = \begin{bmatrix} -1 & 0 \\ 2 & 1 \end{bmatrix}. \tag{28}$$

Let $S = \{(\mathbf{x}, y)\} = \{((1, 2)^{\top}, +1)\}$ be a training set with only one point. By Theorem 4.1, the limit direction $\boldsymbol{\beta}_{\mathrm{steer}}^{\infty}$ of a linear-steerable-network-parameterized linear predictor satisfies

$$\boldsymbol{\beta}_{\mathrm{steer}}^{\infty} \propto \frac{1}{2} \sum_{g \in G} g^{\top} \boldsymbol{\gamma}^* = \begin{bmatrix} 0 & 1 \\ 0 & 1 \end{bmatrix} \boldsymbol{\gamma}^* = \begin{bmatrix} 0 & 1 \\ 0 & 1 \end{bmatrix} \begin{bmatrix} 0 \\ \frac{1}{3} \end{bmatrix} = \begin{bmatrix} \frac{1}{3} \\ \frac{1}{3} \end{bmatrix}, \tag{29}$$

where $\boldsymbol{\gamma}^* = (0, \frac{1}{3})^{\top}$ is the max-margin SVM solution for the transformed data set $\overline{S} = \{(\overline{\mathbf{x}}, y)\} = \{((0, 3)^{\top}, +1)\}$. On the contrary, by Corollary 4.3, the limit direction $\boldsymbol{\beta}_{\mathrm{fc}}^{\infty}$ of a linear fully-connected network trained on the augmented data set $S_{\mathrm{aug}} = \{((1, 2)^{\top}, +1), ((-1, 4)^{\top}, +1)\}$ aligns with the max-margin SVM solution $\boldsymbol{\gamma}_{\mathrm{aug}}^*$ for $S_{\mathrm{aug}}$,

$$\boldsymbol{\beta}_{\mathrm{fc}}^{\infty} \propto \boldsymbol{\gamma}_{\mathrm{aug}}^* = \arg\min_{\boldsymbol{\gamma} \in \mathbb{R}^2} \|\boldsymbol{\gamma}\|^2, \quad \text{s.t. } y \langle \mathbf{x}, \boldsymbol{\gamma} \rangle \geq 1, \forall (\mathbf{x}, y) \in S_{\mathrm{aug}} \tag{30}$$

$$= (0.2, 0.4)^{\top} \not\propto \boldsymbol{\beta}_{\mathrm{steer}}^{\infty}. \tag{31}$$

However, we demonstrate below that the equivalence between linear steerable networks and data augmentation can be re-established for non-unitary $\rho_0$ by defining a new inner product on $\mathbb{R}^{d_0}$,

$$\langle \mathbf{v}, \mathbf{w} \rangle_{\rho_0} := \mathbf{v}^{\top} \left( \frac{1}{|G|} \sum_{g \in G} \rho_0(g)^{\top} \rho_0(g) \right)^{1/2} \mathbf{w}, \quad \forall \mathbf{v}, \mathbf{w} \in \mathcal{X}_0 = \mathbb{R}^{d_0}. \tag{32}$$

When $\rho_0$ is unitary, $\langle \mathbf{v}, \mathbf{w} \rangle_{\rho_0} = \langle \mathbf{v}, \mathbf{w} \rangle$ is the normal Euclidean inner product. With this new inner product, we modify the linear fully-connected network and its empirical loss on the augmented data set $S_{\mathrm{aug}} = \{(g\mathbf{x}_i, y_i) : g \in G, i \in [n]\}$ as

$$f_{\mathrm{fc}}^{\rho_0}(\mathbf{x}; \mathbf{W}) := \langle \mathbf{x}, \mathcal{P}_{\mathrm{fc}}(\mathbf{W}) \rangle_{\rho_0}, \tag{33}$$

$$\mathcal{L}_{\mathcal{P}_{\mathrm{fc}}}^{\rho_0}(\mathbf{W}; S_{\mathrm{aug}}) := \sum_{g \in G} \sum_{i=1}^{n} \ell_{\exp}\left(f_{\mathrm{fc}}^{\rho_0}(g\mathbf{x}_i; \mathbf{W}), y_i\right) = \sum_{g \in G} \sum_{i=1}^{n} \ell_{\exp}\left(\langle g\mathbf{x}_i, \mathcal{P}_{\mathrm{fc}}(\mathbf{W}) \rangle_{\rho_0}, y_i\right). \tag{34}$$

The following corollary shows that, for non-unitary $\rho_0$ on $\mathcal{X}_0$, the implicit bias of a linear steerable network is again the same as that of a modified linear fully-connected network $f_{\mathrm{fc}}^{\rho_0}(\mathbf{x}; \mathbf{W})$ trained under data augmentation.

**Corollary 5.5.** *Let $\boldsymbol{\beta}_{\mathrm{steer}}^{\infty}$ be the same as that in Corollary 5.1. Let $\boldsymbol{\beta}_{\mathrm{fc}}^{\rho_0, \infty} = \lim_{t \to \infty} \frac{\boldsymbol{\beta}_{\mathrm{fc}}^{\rho_0}(t)}{\|\boldsymbol{\beta}_{\mathrm{fc}}^{\rho_0}(t)\|}$ be the limit direction of $\boldsymbol{\beta}_{\mathrm{fc}}^{\rho_0}(t) = \mathcal{P}_{\mathrm{fc}}(\mathbf{W}(t))$ under the gradient flow of the modified empirical loss $\mathcal{L}_{\mathcal{P}_{\mathrm{fc}}}^{\rho_0}(\mathbf{W}; S_{\mathrm{aug}})$ (34) for a linear fully-connected network on the **augmented** data set $S_{\mathrm{aug}}$. Then*

$$\boldsymbol{\beta}_{\mathrm{steer}}^{\infty} \propto \left( \frac{1}{|G|} \sum_{g \in G} \rho_0(g)^{\top} \rho_0(g) \right)^{1/2} \boldsymbol{\beta}_{\mathrm{fc}}^{\rho_0, \infty}. \tag{35}$$

*Consequently, we have $\langle \mathbf{x}, \boldsymbol{\beta}_{\mathrm{steer}}^{\infty} \rangle \propto \langle \mathbf{x}, \boldsymbol{\beta}_{\mathrm{fc}}^{\rho_0, \infty} \rangle_{\rho_0}$ for all $\mathbf{x} \in \mathbb{R}^{d_0}$.*

## 6 Improved margin and generalization

We demonstrate in this section the improved margin and generalization of linear steerable networks over their non-invariant counterparts. In what follows, we assume $\rho_0$ to be unitary.

The following theorem shows that the margin of a linear-steerable-network-parameterized predictor $\boldsymbol{\beta}_{\mathrm{steer}}^{\infty}$ on the augmented data set $S_{\mathrm{aug}}$ is always larger than that of a linear fully-connected network $\boldsymbol{\beta}_{\mathrm{fc}}^{\infty}$, suggesting improved $L^2$-robustness of the steerable-network-parameterized classifier.

**Theorem 6.1.** *Let $\beta_{\text{steer}}^{\infty}$ be the directional limit of a linear-steerable-network-parameterized predictor trained on the **original** data set $S = \{(\mathbf{x}_i, y_i), i \in [n]\}$; let $\beta_{\text{fc}}^{\infty}$ be that of a linear fully-connected network **also** trained on the same data set $S$. Let $M_{\text{steer}}$ and $M_{\text{fc}}$, respectively, be the (signed) margin of $\beta_{\text{steer}}^{\infty}$ and $\beta_{\text{fc}}^{\infty}$ on the **augmented** data set $S_{\text{aug}} = \{(g\mathbf{x}_i, y_i) : i \in [n], g \in G\}$, i.e.,*

$$M_{\text{steer}} := \min_{i \in [n], g \in G} y_i \langle \beta_{\text{steer}}^{\infty}, g\mathbf{x}_i \rangle, \quad M_{\text{fc}} := \min_{i \in [n], g \in G} y_i \langle \beta_{\text{fc}}^{\infty}, g\mathbf{x}_i \rangle. \tag{36}$$

*Then we always have $M_{\text{steer}} \geq M_{\text{fc}}$.*

Finally, we aim to quantify the improved generalization of linear steerable networks compared to fully-connected networks in binary classification of *linearly separable* group-invariant distributions defined below.

**Definition 6.2.** *A distribution $\mathcal{D}$ on $\mathbb{R}^{d_0} \times \{\pm 1\}$ is called linearly separable if there exists $\beta \in \mathbb{R}^{d_0}$ such that*

$$\mathbb{P}_{(\mathbf{x},y) \sim \mathcal{D}} \left[ y \langle \mathbf{x}, \beta \rangle \geq 1 \right] = 1. \tag{37}$$

It is easy to verify (by Lemma F.1) that if $\mathcal{D}$ is $G$-invariant and linearly separable, then $\mathcal{D}$ can be separated by a $G$-invariant linear classifier. The following theorem establishes the generalization bound of linear steerable networks in separable group-invariant binary classification.

**Theorem 6.3.** *Let $\mathcal{D}$ be a $G$-invariant distribution over $\mathbb{R}^{d_0} \times \{\pm 1\}$ that is linearly separable by an invariant classifier $\beta_0 \in \mathbb{R}_G^{d_0}$. Define*

$$\overline{R} = \inf \left\{ r > 0 : \|\overline{\mathbf{x}}\| \leq r \text{ with probability } 1 \right\}. \tag{38}$$

*Let $S = \{(\mathbf{x}_i, y_i)\}_{i=1}^{n}$ be i.i.d. samples from $\mathcal{D}$, and let $\beta_{\text{steer}}^{\infty}$ be the limit direction of a steerable-network-parameterized linear predictor trained using gradient flow on $S$. Then, for any $\delta > 0$, we have with probability at least $1 - \delta$ (over random samples $S \sim \mathcal{D}^n$) that*

$$\mathbb{P}_{(\mathbf{x},y) \sim \mathcal{D}} \left[ y \neq \text{sign} \left( \langle \mathbf{x}, \beta_{\text{steer}}^{\infty} \rangle \right) \right] \leq \frac{2\overline{R} \|\beta_0\|}{\sqrt{n}} + \sqrt{\frac{\log(1/\delta)}{2n}}. \tag{39}$$

**Remark 6.4.** *In comparison, let $\beta_{\text{fc}}^{\infty}$ be the limit direction of a fully-connected-network-parameterized linear predictor trained on $S$. Then with probability at least $1 - \delta$, we have*

$$\mathbb{P}_{(\mathbf{x},y) \sim \mathcal{D}} \left[ y \neq \text{sign} \left( \langle \mathbf{x}, \beta_{\text{fc}}^{\infty} \rangle \right) \right] \leq \frac{2R \|\beta_0\|}{\sqrt{n}} + \sqrt{\frac{\log(1/\delta)}{2n}}, \tag{40}$$

*where $R = \inf \left\{ r > 0 : \|\mathbf{x}\| \leq r \text{ with probability } 1 \right\}$. This is the classical generalization result for max-margin SVM (see, e.g., Shalev-Shwartz and Ben-David [2014].) Eq. (40) can also be viewed as a special case of Eq. (39), as a fully-connected network is a G-CNN with $G = \{e\}$ (cf. Remark 4.2), and therefore $\overline{\mathbf{x}} = \mathbf{x}$ and $R = \overline{R}$.*

*By Proposition 3.6, the map $\mathbf{x} \rightarrow \overline{\mathbf{x}}$ is an orthogonal projection, and thus we always have $\overline{R} \leq R$. Therefore the generalization bound for steerable networks in (39) is always smaller than that of the fully-connected network in (40).*

**Remark 6.5.** *A comparison between Eq. (39) and Eq. (40) reveals that the improved generalization of linear steerable network does not necessarily depend on the group size $|G|$. Instead, it depends on how far the distribution $\mathcal{D}$'s support is from the subspace $\mathbb{R}_G^{d_0}$, such that $\overline{R}$ could be much smaller than $R$. In fact, if the support of $\mathcal{D}$ is contained in $R_G^{d_0}$, then $\overline{R} = R$, and the steerable network does not achieve any generalization gain. This is consistent with Theorem 4.1, as in this case, the transformed data set $\overline{S} = \{(\overline{\mathbf{x}}_i, y_i) : i \in [n]\}$ is the same as the original data set $S$.*

## 7 Conclusion and future work

In this work, we analyzed the implicit bias of gradient flow on general linear group-equivariant steerable networks in group-invariant binary classification. Our findings indicate that the parameterized predictor converges in a direction that aligns with the unique group-invariant classifier with a maximum margin that is dependent on the input representation. As a corollary of our main result,

we established the equivalence between data augmentation and learning with steerable networks in our setting. Finally, we demonstrated that linear steerable networks outperform their non-invariant counterparts in terms of improved margin and generalization bound.

A limitation of our result is that the implicit bias of gradient flow studied herein holds in an asymptotic sense, and the convergence rate to the directional limit might be extremely slow. This is consistent with the findings in, *e.g.*, [Soudry et al., 2018, Yun et al., 2021]. Understanding the behavior of gradient flow in a non-asymptotic regime is an important direction for future work. Furthermore, in our current framework, we assume that the first-layer equivariant map is represented by the $G$-lifting map. This assumption ensures that the linear steerable network possesses sufficient capacity to parameterize all $G$-invariant linear classifiers. Exploring the implicit bias of steerable networks without this assumption would be a compelling next step. The removal of this constraint could facilitate the generalization of our findings from finite groups to compact groups.

## Acknowledgements

The research was partially supported by NSF under DMS-2052525, DMS-2140982, and DMS-2244976.

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

# A G-CNNs

A special case of the steerable networks is the *group convolutional neural network* (G-CNN), wherein equivariance is achieved through *group convolutions*. More specifically, for each $l \in [L-1]$, the hidden representation space $(\mathcal{X}_l, \rho_l)$ is set to

$$\mathcal{X}_l = (\mathbb{R}^{d_l})^G = \{\mathbf{x}_l : G \to \mathbb{R}^{d_l}\}, \quad \rho_l(g)\mathbf{x}_l(h) := \mathbf{x}_l(g^{-1}h) \in \mathbb{R}^{d_l}, \forall g, h \in G. \tag{41}$$

The representation $\rho_l \in \mathrm{GL}(\mathcal{X}_l)$ in (41) is known as the *regular representation* of $G$. Intuitively, $\mathbf{x}_l \in \mathcal{X}_l$ can be viewed as a matrix of size $d_l \times |G|$, and $\rho_l(g)$ is a permutation of the columns of $\mathbf{x}_l$. With this choice of $\{(\mathcal{X}_l, \rho_l)\}_{l=0}^L$, a (linear) G-CNN is built as follows.

First layer: the first-layer equivariant map $\Psi_1^{\text{G-CNN}} : \mathcal{X}_0 \to \mathcal{X}_1$ of a G-CNN is defined as

$$\Psi_1^{\text{G-CNN}}(\mathbf{x}; \mathbf{w}_1)(g) = \mathbf{w}_1^\top g^{-1}\mathbf{x} \in \mathbb{R}^{d_1}, \quad \forall \mathbf{x} \in \mathbb{R}^{d_0}, \tag{42}$$

where $\mathbf{w}_1 = (w_1^{j,k})_{j,k} \in \mathbb{R}^{d_0 \times d_1}$ are the trainable parameters of the first layer. Eq. (42) is called a *G-lifting map* as it lifts a Euclidean signal $\mathbf{x} \in \mathbb{R}^{d_0}$ to a function $\Psi_1^{\text{G-CNN}}(\mathbf{x}; \mathbf{w}_1)$ on $G$.

Hidden layers: For $l \in \{2, \cdots, L-1\}$, define $\Psi_l^{\text{G-CNN}} : \mathcal{X}_{l-1} \to \mathcal{X}_l$ as the *group convolutions*,

$$\Psi_l^{\text{G-CNN}}(\mathbf{x}; \mathbf{w}_l)(g) = \sum_{h \in G} \mathbf{w}_l^\top (h^{-1}g)\mathbf{x}(h) \in \mathbb{R}^{d_l}, \quad \forall \mathbf{x} \in \mathcal{X}_{l-1}, \tag{43}$$

where $\mathbf{w}_l(g) \in \mathbb{R}^{d_{l-1} \times d_l}, \forall g \in G$. Equivalently, $\mathbf{w}_l \in \mathbb{R}^{d_{l-1} \times d_l \times |G|}$ can be viewed as a 3D tensor.

Last layer: Define $\Psi_L^{\text{G-CNN}} : \mathcal{X}_{L-1} \to \mathcal{X}_L$ as the *group-pooling* followed by a fully-connected layer,

$$\Psi_L^{\text{G-CNN}}(\mathbf{x}; \mathbf{w}_L) = \mathbf{w}_L^\top \frac{1}{|G|} \sum_{g \in G} \mathbf{x}(g), \quad \forall \mathbf{x} \in \mathcal{X}_{L-1}, \tag{44}$$

where $\mathbf{w}_L \in \mathbb{R}^{d_{L-1}}$ is the weight of the last layer. An $L$-layer linear G-CNN is then the composition

$$f_{\text{G-CNN}}(\mathbf{x}; \mathbf{W}) = \Psi_L^{\text{G-CNN}}(\cdots \Psi_2^{\text{G-CNN}}(\Psi_1^{\text{G-CNN}}(\mathbf{x}; \mathbf{w}_1); \mathbf{w}_2) \cdots ; \mathbf{w}_L), \quad \mathbf{W} = [\mathbf{w}_l]_{l=1}^L, \tag{45}$$

where $\mathbf{W} \in \mathbb{R}^{d_0 \times d_1} \times \left[ \prod_{l=2}^{L-1} \mathbb{R}^{d_{l-1} \times d_l \times |G|} \right] \times \mathbb{R}^{d_{L-1}} =: \mathcal{W}_{\text{G-CNN}}$ are the trainable weights.

**Remark A.1.** *As a special case of linear steerable networks, a linear G-CNN $f_{\text{G-CNN}}(\mathbf{x}; \mathbf{W})$ (45) can be written as*

$$f_{\text{G-CNN}}(\mathbf{x}; \mathbf{W}) = \langle \mathbf{x}, \mathcal{P}_{\text{G-CNN}}(\mathbf{W}) \rangle = \langle \overline{\mathbf{x}}, \overline{\mathcal{P}}_{\text{G-CNN}}(\mathbf{W}) \rangle, \tag{46}$$

*where*

$$\overline{\mathcal{P}}_{\text{G-CNN}}(\mathbf{W}) := \mathbf{w}_1 \left[ \prod_{l=2}^{L-1} \left( \sum_{g \in G} \mathbf{w}_l(g) \right) \right] \mathbf{w}_L, \quad \mathcal{P}_{\text{G-CNN}}(\mathbf{W}) := \left[ \frac{1}{|G|} \sum_{g \in G} g^\top \right] \overline{\mathcal{P}}_{\text{G-CNN}}(\mathbf{W}) \tag{47}$$

# B Proofs in Section 3

**Proposition 3.3.** *Let $f_{\text{steer}}(\mathbf{x}; \mathbf{W})$ be the linear steerable network satisfying Assumption 3.1, where $\mathbf{W} = [\mathbf{w}_l]_{l=1}^L \in \mathcal{W}_{\text{steer}}$ is the collection of all model parameters. There exists a multi-linear map $M : (\mathbf{w}_2, \cdots, \mathbf{w}_L) \mapsto M(\mathbf{w}_2, \cdots, \mathbf{w}_L) \in \mathbb{R}^{d_1}$ such that for all $\mathbf{x} \in \mathbb{R}^{d_0}$ and $\mathbf{W} \in \mathcal{W}_{\text{steer}}$,*

$$f_{\text{steer}}(\mathbf{x}; \mathbf{W}) = f_{\text{steer}}(\overline{\mathbf{x}}; \mathbf{W}) = \langle \overline{\mathbf{x}}, \mathbf{w}_1 M(\mathbf{w}_2, \cdots, \mathbf{w}_L) \rangle, \tag{48}$$

*where $\overline{\mathbf{x}} := \frac{1}{|G|} \sum_{g \in G} g\mathbf{x}$ is the average of all elements on the group orbit of $\mathbf{x} \in \mathbb{R}^{d_0}$.*

*Proof.* Since the linear steerable network $f_{\text{steer}}(\mathbf{x}; \mathbf{W})$ is $G$-invariant, we have $f_{\text{steer}}(\mathbf{x}; \mathbf{w}) = f_{\text{steer}}(g\mathbf{x}; \mathbf{w})$ for all $g \in G$. Therefore,

$$f_{\text{steer}}(\mathbf{x}; \mathbf{W}) = \frac{1}{|G|} \sum_{g \in G} f_{\text{steer}}(g\mathbf{x}; \mathbf{W}) = f_{\text{steer}} \left( \frac{1}{|G|} \sum_{g \in G} g\mathbf{x}; \mathbf{w} \right) = f_{\text{steer}}(\overline{\mathbf{x}}; \mathbf{W}), \tag{49}$$

where the second equality is due to the linearity of $f_{\text{steer}}(\mathbf{x}; \mathbf{w})$ in $\mathbf{x}$. We thus have

$$f_{\text{steer}}(\mathbf{x}; \mathbf{W}) = f_{\text{steer}}(\overline{\mathbf{x}}; \mathbf{W}) = \Psi_L^{\text{steer}}(\cdots \Psi_2^{\text{steer}}(\Psi_1^{\text{steer}}(\overline{\mathbf{x}}; \mathbf{w}_1); \mathbf{w}_2) \cdots ; \mathbf{w}_L) \tag{50}$$

$$= \left\langle \Psi_1^{\text{steer}}(\overline{\mathbf{x}}; \mathbf{w}), \Phi(\mathbf{w}_2, \cdots, \mathbf{w}_L) \right\rangle, \tag{51}$$

where $\Phi(\mathbf{w}_2, \cdots, \mathbf{w}_L) \in \mathcal{X}_1 = (\mathbb{R}^{d_1})^G$ is multi-linear with respect to $(\mathbf{w}_1, \cdots, \mathbf{w}_L)$. Under Assumption 3.1, we have

$$f_{\text{steer}}(\mathbf{x}; \mathbf{W}) = \sum_{g \in G} \left\langle \Psi_1^{\text{steer}}(\overline{\mathbf{x}}; \mathbf{w})(g), \Phi(\mathbf{w}_2, \cdots, \mathbf{w}_L)(g) \right\rangle \tag{52}$$

$$= \sum_{g \in G} \left\langle \mathbf{w}_1^\top g^{-1} \overline{\mathbf{x}}, \Phi(\mathbf{w}_2, \cdots, \mathbf{w}_L)(g) \right\rangle \tag{53}$$

$$= \sum_{g \in G} \left\langle \mathbf{w}_1^\top \overline{\mathbf{x}}, \Phi(\mathbf{w}_2, \cdots, \mathbf{w}_L)(g) \right\rangle \tag{54}$$

$$= \left\langle \overline{\mathbf{x}}, \mathbf{w}_1 \left[ \sum_{g \in G} \Phi(\mathbf{w}_2, \cdots, \mathbf{w}_L)(g) \right] \right\rangle \tag{55}$$

$$= \left\langle \overline{\mathbf{x}}, \mathbf{w}_1 M(\mathbf{w}_2, \cdots, \mathbf{w}_L) \right\rangle, \tag{56}$$

where $M(\mathbf{w}_2, \cdots, \mathbf{w}_L) := \sum_{g \in G} \Phi(\mathbf{w}_2, \cdots, \mathbf{w}_L)(g)$ is also multi-linear in $(\mathbf{w}_2, \cdots, \mathbf{w}_L)$. $\square$

**Proposition 3.6.** *Let $\mathbb{R}_G^{d_0} \subset \mathbb{R}^{d_0}$ be the subspace of $G$-invariant linear predictors, i.e., $\mathbb{R}_G^{d_0} = \left\{ \boldsymbol{\beta} \in \mathbb{R}^{d_0} : \boldsymbol{\beta}^\top \mathbf{x} = \boldsymbol{\beta}^\top g\mathbf{x}, \forall \mathbf{x} \in \mathbb{R}^{d_0}, \forall g \in G \right\}$. Then*

*(a) $\mathbb{R}_G^{d_0}$ is characterized by*

$$\mathbb{R}_G^{d_0} = \bigcap_{g \in G} \ker(I - g^\top) = \text{Range} \left( \frac{1}{|G|} \sum_{g \in G} g^\top \right). \tag{57}$$

*(b) Let $\mathcal{A} : \mathbb{R}^{d_0} \to \mathbb{R}^{d_0}$ be the group-averaging map,*

$$\mathcal{A}(\boldsymbol{\beta}) := \overline{\boldsymbol{\beta}} = \frac{1}{|G|} \sum_{g \in G} g\boldsymbol{\beta}. \tag{58}$$

*Then its adjoint $\mathcal{A}^\top : \boldsymbol{\beta} \mapsto \frac{1}{|G|} \sum_{g \in G} g^\top \boldsymbol{\beta}$ is a projection operator from $\mathbb{R}^{d_0}$ to $\mathbb{R}_G^{d_0}$. In other words, $\text{Range}(\mathcal{A}^\top) = \mathbb{R}_G^{d_0}$ and $\mathcal{A}^\top \circ \mathcal{A}^\top = \mathcal{A}^\top$.*

*(c) If $G$ acts unitarily on $\mathcal{X}_0$, i.e., $\rho_0(g^{-1}) = \rho_0(g)^\top$, then $\mathcal{A} = \mathcal{A}^\top$ is self-adjoint. This implies that $\mathcal{A} : \boldsymbol{\beta} \mapsto \overline{\boldsymbol{\beta}}$ is an orthogonal projection from $\mathbb{R}^{d_0}$ onto $\mathbb{R}_G^{d_0}$. In particular, we have*

$$\overline{\boldsymbol{\beta}} = \boldsymbol{\beta} \iff \boldsymbol{\beta} \in \mathbb{R}_G^{d_0}, \quad \text{and } \|\overline{\boldsymbol{\beta}}\| \le \|\boldsymbol{\beta}\|, \forall \boldsymbol{\beta} \in \mathbb{R}^{d_0}. \tag{59}$$

*Proof.* To prove (a), for the first equality, a vector $\boldsymbol{\beta} \in \mathbb{R}_G^{d_0}$ if and only if $0 = \boldsymbol{\beta}^\top (I - g)\mathbf{x} = \left\langle (I - g)^\top \boldsymbol{\beta}, \mathbf{x} \right\rangle, \forall \mathbf{x} \in \mathbb{R}^{d_0}, \forall g \in G$. This is equivalent to $\boldsymbol{\beta} \in \bigcap_{g \in G} \ker(I - g^\top)$, and therefore $\mathbb{R}_G^{d_0} = \bigcap_{g \in G} \ker(I - g^\top)$.

For the second equality, if $\boldsymbol{\beta} \in \bigcap_{g \in G} \ker(I - g^\top)$, then $(I - g^\top)\boldsymbol{\beta} = 0, \forall g \in G$. Hence

$$0 = \frac{1}{|G|} \sum_{g \in G} (I - g^\top)\boldsymbol{\beta} = \boldsymbol{\beta} - \frac{1}{|G|} \sum_{g \in G} g^\top \boldsymbol{\beta} \tag{60}$$

$$\implies \boldsymbol{\beta} = \frac{1}{|G|} \sum_{g \in G} g^\top \boldsymbol{\beta} \in \text{Range} \left( \frac{1}{|G|} \sum_{g \in G} g^\top \right) \tag{61}$$

On the other hand, if $\boldsymbol{\beta} = \frac{1}{|G|} \sum_{h \in G} h^\top \mathbf{y}$ for some $\mathbf{y} \in \mathbb{R}^{d_0}$, then for all $g \in G$,

$$\left(I - g^\top\right) \boldsymbol{\beta} = \left(I - g^\top\right) \frac{1}{|G|} \sum_{h \in G} h^\top \mathbf{y} = \frac{1}{|G|} \sum_{h \in G} h^\top \mathbf{y} - \frac{1}{|G|} \sum_{h \in G} (hg)^\top \mathbf{y} = 0. \qquad (62)$$

Thus $\boldsymbol{\beta} \in \bigcap_{g \in G} \ker(I - g^\top)$.

Point (b) can be easily derived from (57) and (60).

To prove (c), notice that

$$\mathcal{A}^\top = \frac{1}{|G|} \sum_{g \in G} g^\top = \frac{1}{|G|} \sum_{g \in G} g^{-1} = \frac{1}{|G|} \sum_{g \in G} g = \mathcal{A}. \qquad (63)$$

Hence $\mathcal{A} = \mathcal{A}^\top$ is self-adjoint. This combined with point (b) implies that $\mathcal{A} = \mathcal{A}^\top : \boldsymbol{\beta} \mapsto \overline{\boldsymbol{\beta}}$ is an orthogonal projection from $\mathbb{R}^{d_0}$ onto $\mathbb{R}_G^{d_0}$. $\qquad\square$

## C  Proof of Theorem 4.1

**Theorem 4.1.** *Under Assumption 3.1 and Assumption 3.7, let $\boldsymbol{\beta}_{\text{steer}}(t) = \mathcal{P}_{\text{steer}}(\mathbf{W}(t))$ be the time-evolution of the $G$-invariant linear predictors parameterized by a linear steerable network trained with gradient flow on the data set $S = \{(\mathbf{x}_i, y_i) : i \in [n]\}$; cf. Eq. (21). Then*

*(a) The directional limit $\boldsymbol{\beta}_{\text{steer}}^\infty = \lim_{t \to \infty} \frac{\boldsymbol{\beta}_{\text{steer}}(t)}{\|\boldsymbol{\beta}_{\text{steer}}(t)\|}$ exists and $\boldsymbol{\beta}_{\text{steer}}^\infty \propto \frac{1}{|G|} \sum_{g \in G} g^\top \boldsymbol{\gamma}^*$, where $\boldsymbol{\gamma}^*$ is the max-$L^2$-margin SVM solution for the **transformed** data $\overline{S} = \{(\overline{\mathbf{x}}_i, y_i) : i \in [n]\}$:*

$$\boldsymbol{\gamma}^* = \arg \min_{\boldsymbol{\gamma} \in \mathbb{R}^{d_0}} \|\boldsymbol{\gamma}\|^2, \quad \text{s.t. } y_i \langle \overline{\mathbf{x}}_i, \boldsymbol{\gamma} \rangle \geq 1, \forall i \in [n]. \qquad (64)$$

*Furthermore, if $G$ acts unitarily on the input space $\mathcal{X}_0$, i.e., $g^{-1} = g^\top$, then $\boldsymbol{\beta}_{\text{steer}}^\infty \propto \boldsymbol{\gamma}^*$.*

*(b) Equivalently, $\boldsymbol{\beta}_{\text{steer}}^\infty$ is proportional to the unique minimizer $\boldsymbol{\beta}^*$ of the problem*

$$\boldsymbol{\beta}^* = \arg \min_{\boldsymbol{\beta} \in \mathbb{R}^{d_0}} \|\text{Proj}_{\text{Range}(\mathcal{A})} \boldsymbol{\beta}\|^2, \quad \text{s.t. } \boldsymbol{\beta} \in \mathbb{R}_G^{d_0}, \text{ and } y_i \langle \mathbf{x}_i, \boldsymbol{\beta} \rangle \geq 1, \forall i \in [n], \qquad (65)$$

*where $\text{Proj}_{\text{Range}(\mathcal{A})}$ is the projection from $\mathbb{R}^{d_0}$ to $\text{Range}(\mathcal{A}) = \text{Range}\left(\frac{1}{|G|} \sum_{g \in G} g\right)$. Moreover, if $G$ acts unitarily on $\mathcal{X}_0$, then*

$$\boldsymbol{\beta}_{\text{steer}}^\infty \propto \boldsymbol{\beta}^* = \arg \min_{\boldsymbol{\beta} \in \mathbb{R}^{d_0}} \|\boldsymbol{\beta}\|^2, \quad \text{s.t. } \boldsymbol{\beta} \in \mathbb{R}_G^{d_0}, \text{ and } y_i \langle \mathbf{x}_i, \boldsymbol{\beta} \rangle \geq 1, \forall i \in [n]. \qquad (66)$$

*Namely, $\boldsymbol{\beta}_{\text{steer}}^\infty$ achieves the maximum $L^2$-margin among all $G$-invariant linear predictors.*

Before proving Theorem 4.1, we need the following lemma which holds for general $L$-homogeneous networks, of which linear steerable networks are a special case. Note that in what follows, $\|\cdot\|$ always denotes the Euclidean norm of a tensor viewed as a one-dimensional vector.

**Lemma C.1** (paraphrased from Lyu and Li [2020] and Ji and Telgarsky [2020]). *Under Assumption 3.7, we have the following results of directional convergence and alignment.*

*(a) $\mathcal{L}_{\mathcal{P}_{\text{steer}}}(\mathbf{W}(t); S) \to 0$, as $t \to \infty$. Consequently, $\|\mathbf{W}(t)\| \to \infty$ and $\|\mathcal{P}_{\text{steer}}(\mathbf{W}(t))\| \to \infty$.*

*(b) The directional convergence and alignment of the parameters $\mathbf{W}(t)$ and the gradients $\nabla_{\mathbf{W}} \mathcal{L}_{\mathcal{P}_{\text{steer}}}(\mathbf{W}(t))$,*

$$\lim_{t \to \infty} \frac{\mathbf{W}(t)}{\|\mathbf{W}(t)\|} = \mathbf{W}^\infty = -\lim_{t \to \infty} \frac{\nabla_{\mathbf{W}} \mathcal{L}_{\mathcal{P}_{\text{steer}}}(\mathbf{W}(t); S)}{\|\nabla_{\mathbf{W}} \mathcal{L}_{\mathcal{P}_{\text{steer}}}(\mathbf{W}(t); S)\|}, \qquad (67)$$

*for some $\mathbf{W}^\infty \in \mathcal{W}_{\text{steer}}$ with $\|\mathbf{W}^\infty\| = 1$.*

*(c) The limit $\mathbf{W}^\infty$ is along the direction of a first-order stationary point of the constrained optimization problem*

$$\min_{\mathbf{W}} \|\mathbf{W}\|^2, \quad \text{s.t. } y_i \langle \mathbf{x}_i, \mathcal{P}_{\text{steer}}(\mathbf{W}) \rangle \geq 1, \forall i \in [n]. \qquad (68)$$

*In other words, there exists a scaling factor $\tau > 0$ such that $\tau \mathbf{W}^\infty$ satisfies the Karush-Kuhn-Tucker (KKT) conditions of (68).*

*Proof of Theorem 4.1.* The first part of the proof is inspired by Gunasekar et al. [2018b]. **To prove (a)**, let $\mathbf{W}^\infty = [\mathbf{w}_l^\infty]_{l=1}^L = \lim_{t\to\infty} \frac{\mathbf{W}(t)}{\|\mathbf{W}(t)\|}$ be the limit direction of $\mathbf{W}(t)$, and let $\widetilde{\mathbf{W}}^\infty = [\widetilde{\mathbf{w}}_l^\infty]_{l=1}^L = \tau\mathbf{W}^\infty$, $\tau > 0$, be the stationary point of problem (68) by Lemma C.1. The KKT condition for $\widetilde{\mathbf{W}}^\infty$ implies the existence of dual variables $\alpha_i \geq 0$, $i \in [n]$, such that

- *Primal feasibility:*

$$y_i \left\langle \mathcal{P}_{\text{steer}}(\widetilde{\mathbf{W}}^\infty), \mathbf{x}_i \right\rangle \geq 1, \quad \forall i \in [n]. \tag{69}$$

- *Stationarity:*

$$\widetilde{\mathbf{W}}^\infty = \nabla_{\mathbf{W}} \mathcal{P}_{\text{steer}}(\widetilde{\mathbf{W}}^\infty) \cdot \left( \sum_{i=1}^n \alpha_i y_i \mathbf{x}_i \right) \tag{70}$$

- *Complementary slackness*:

$$y_i \left\langle \mathcal{P}_{\text{steer}}(\widetilde{\mathbf{W}}^\infty), \mathbf{x}_i \right\rangle > 1 \implies \alpha_i = 0. \tag{71}$$

We claim that $\boldsymbol{\gamma}^* = \overline{\mathcal{P}}_{\text{steer}}(\widetilde{\mathbf{W}}^\infty)$ is a stationary point of the following problem

$$\min_{\boldsymbol{\gamma} \in \mathbb{R}^{d_0}} \|\boldsymbol{\gamma}\|^2, \quad \text{s.t. } y_i \langle \overline{\mathbf{x}}_i, \boldsymbol{\gamma} \rangle \geq 1, \forall i \in [n]. \tag{72}$$

That is, there exists $\tilde{\alpha}_i \geq 0$, $i \in [n]$, such that the following conditions are satisfied.

- *Primal feasibility:*

$$y_i \left\langle \overline{\mathcal{P}}_{\text{steer}}(\widetilde{\mathbf{W}}^\infty), \overline{\mathbf{x}}_i \right\rangle \geq 1, \quad \forall i \in [n]. \tag{73}$$

- *Stationarity:*

$$\overline{\mathcal{P}}_{\text{steer}}(\widetilde{\mathbf{W}}^\infty) = \sum_{i=1}^n \tilde{\alpha}_i y_i \overline{\mathbf{x}}_i \tag{74}$$

- *Complementary slackness*:

$$y_i \left\langle \overline{\mathcal{P}}_{\text{steer}}(\widetilde{\mathbf{W}}^\infty), \overline{\mathbf{x}}_i \right\rangle > 1 \implies \tilde{\alpha}_i = 0. \tag{75}$$

Indeed, Eq. (73) holds due to Eq. (69) and $\left\langle \mathcal{P}_{\text{steer}}(\widetilde{\mathbf{W}}^\infty), \mathbf{x}_i \right\rangle = \left\langle \overline{\mathcal{P}}_{\text{steer}}(\widetilde{\mathbf{W}}^\infty), \overline{\mathbf{x}}_i \right\rangle$. Moreover, by the definition of $\mathcal{P}_{\text{steer}}(\mathbf{W})$ and $\overline{\mathcal{P}}_{\text{steer}}(\mathbf{W})$ in (10), we have for any $\mathbf{z} \in \mathbb{R}^{d_0}$,

$$\nabla_{\mathbf{w}_1} \overline{\mathcal{P}}_{\text{steer}}(\mathbf{W}) \cdot \mathbf{z} = \mathbf{z}[M(\mathbf{w}_2, \cdots, \mathbf{w}_L)]^\top \tag{76}$$

$$\nabla_{\mathbf{w}_1} \mathcal{P}_{\text{steer}}(\mathbf{W}) \cdot \mathbf{z} = \left( \frac{1}{|G|} \sum_{g \in G} g \right) \mathbf{z}[M(\mathbf{w}_2, \cdots, \mathbf{w}_L)]^\top = \overline{\mathbf{z}}[M(\mathbf{w}_2, \cdots, \mathbf{w}_L)]^\top. \tag{77}$$

Therefore Eq. (70) implies

$$\widetilde{\mathbf{w}}_1^\infty = \nabla_{\mathbf{w}_1} \mathcal{P}_{\text{steer}}(\widetilde{\mathbf{W}}^\infty) \cdot \left( \sum_{i=1}^n \alpha_i y_i \mathbf{x}_i \right) = \left( \sum_{i=1}^n \alpha_i y_i \overline{\mathbf{x}}_i \right) \cdot [M(\widetilde{\mathbf{w}}_2^\infty, \cdots, \widetilde{\mathbf{w}}_L^\infty)]^\top \tag{78}$$

Hence

$$\overline{\mathcal{P}}_{\text{steer}}(\widetilde{\mathbf{W}}^\infty) = \widetilde{\mathbf{w}}_1^\infty M(\widetilde{\mathbf{w}}_2^\infty, \cdots, \widetilde{\mathbf{w}}_L^\infty) = \left( \sum_{i=1}^n \alpha_i y_i \overline{\mathbf{x}}_i \right) \|M(\widetilde{\mathbf{w}}_2^\infty, \cdots, \widetilde{\mathbf{w}}_L^\infty)\|^2$$

$$= c \sum_{i=1}^n \alpha_i y_i \overline{\mathbf{x}}_i, \tag{79}$$

where $c = \|M(\widetilde{\mathbf{w}}_2^\infty, \cdots, \widetilde{\mathbf{w}}_L^\infty)\|^2 \geq 0$. From (73) we know that $\overline{\mathcal{P}}_{\text{steer}}(\widetilde{\mathbf{W}}^\infty) \neq 0$ and hence $c > 0$. Let $\tilde{\alpha}_i = c\alpha_i, \forall i \in [n]$. Then $\tilde{\alpha}_i \geq 0$, and the stationarity (74) is satisfied due to Eq. (79), and the complementary slackness also holds due to $\tilde{\alpha}_i$ being a positive scaling of $\alpha_i$ for all $i \in [n]$. Therefore $\boldsymbol{\gamma}^* = \overline{\mathcal{P}}_{\text{steer}}(\widetilde{\mathbf{W}}^\infty)$ is a stationary point of (72). Since problem (72) is strongly convex, $\boldsymbol{\gamma}^* = \overline{\mathcal{P}}_{\text{steer}}(\widetilde{\mathbf{W}}^\infty)$ is in fact the unique minimizer of (72). Hence the limit direction of the predictor $\mathcal{P}_{\text{steer}}(\mathbf{W}(t))$ is

$$\boldsymbol{\beta}_{\text{steer}}^\infty = \lim_{t \to \infty} \frac{\mathcal{P}_{\text{steer}}(\mathbf{W}(t))}{\|\mathcal{P}_{\text{steer}}(\mathbf{W}(t))\|} = \lim_{t \to \infty} \frac{\left(\frac{1}{|G|}\sum_{g \in G} g^\top\right)\overline{\mathcal{P}}_{\text{steer}}(\mathbf{W}(t))}{\left\|\left(\frac{1}{|G|}\sum_{g \in G} g^\top\right)\overline{\mathcal{P}}_{\text{steer}}(\mathbf{W}(t))\right\|} \tag{80}$$

$$= \lim_{t \to \infty} \frac{\left(\frac{1}{|G|}\sum_{g \in G} g^\top\right)\overline{\mathcal{P}}_{\text{steer}}\left(\frac{\tau\mathbf{W}(t)}{\|\mathbf{W}(t)\|}\right)}{\left\|\left(\frac{1}{|G|}\sum_{g \in G} g^\top\right)\overline{\mathcal{P}}_{\text{steer}}\left(\frac{\tau\mathbf{W}(t)}{\|\mathbf{W}(t)\|}\right)\right\|} = \frac{\left(\frac{1}{|G|}\sum_{g \in G} g^\top\right)\overline{\mathcal{P}}_{\text{steer}}\left(\widetilde{\mathbf{W}}^\infty\right)}{\left\|\left(\frac{1}{|G|}\sum_{g \in G} g^\top\right)\overline{\mathcal{P}}_{\text{steer}}\left(\widetilde{\mathbf{W}}^\infty\right)\right\|} \tag{81}$$

$$\propto \frac{1}{|G|}\sum_{g \in G} g^\top\boldsymbol{\gamma}^*, \tag{82}$$

where the third equality is due to $\overline{\mathcal{P}}_{\text{steer}}$ being $L$-homogeneous (12), and the fourth equality comes from the continuity of $\overline{\mathcal{P}}_{\text{steer}}$ and that $\mathcal{P}_{\text{steer}}(\widetilde{\mathbf{W}}^\infty) = \left(\frac{1}{|G|}\sum_{g \in G} g^\top\right)\overline{\mathcal{P}}_{\text{steer}}\left(\widetilde{\mathbf{W}}^\infty\right) \neq 0$ (because otherwise (69) can not hold.)

Finally, if $G$ acts unitarily on $\mathcal{X}_0$, then Eq. (74) combined with Proposition 3.6 implies that $\boldsymbol{\gamma}^* = \overline{\mathcal{P}}_{\text{steer}}(\widetilde{\mathbf{W}}^\infty) \in \mathbb{R}_G^{d_0}$. Hence

$$\boldsymbol{\beta}_{\text{steer}}^\infty \propto \frac{1}{|G|}\sum_{g \in G} g^\top\boldsymbol{\gamma}^* = \frac{1}{|G|}\sum_{g \in G} g\boldsymbol{\gamma}^* = \overline{\boldsymbol{\gamma}^*} = \boldsymbol{\gamma}^*, \tag{83}$$

where the last equality is again due to Proposition 3.6. This concludes the proof of (a).

**To prove (b)**, we first show that problem (65) has a unique minimizer. To prove this, notice that $\text{Proj}_{\text{Range}(\mathcal{A})} : \mathbb{R}_G^{d_0} \to \text{Range}(\mathcal{A})$ is injective; indeed, for any $\boldsymbol{\beta} \in \mathbb{R}_G^{d_0}$,

$$\text{Proj}_{\text{Range}(\mathcal{A})}\boldsymbol{\beta} = 0 \implies \boldsymbol{\beta} \in \text{Range}(\mathcal{A})^\perp = \ker(\mathcal{A}^\top). \tag{84}$$

Thus $0 = \mathcal{A}^\top\boldsymbol{\beta} = \boldsymbol{\beta}$, where the second equality is due to $\mathcal{A}^\top$ being a projection onto $\text{Range}(\mathcal{A}^\top) = \mathbb{R}_G^{d_0}$ (Proposition 3.6) and $\boldsymbol{\beta} \in \mathbb{R}_G^{d_0}$. Therefore the objective function in (65) is strongly convex on $\mathbb{R}_G^{d_0}$ and there exists a unique minimizer.

To show $\boldsymbol{\beta}_{\text{steer}}^\infty \propto \boldsymbol{\beta}^*$, it suffices to verify that $\frac{1}{|G|}\sum_{g \in G} g^\top\boldsymbol{\gamma}^* = \boldsymbol{\beta}^*$ is the minimizer of (65). To this end, we notice that

- $\frac{1}{|G|}\sum_{g \in G} g^\top\boldsymbol{\gamma}^* = \mathcal{A}^\top\boldsymbol{\gamma}^* \in \mathbb{R}_G^{d_0}$ by Proposition 3.6;

- for all $i \in [n]$

$$y_i\left\langle \mathbf{x}_i, \frac{1}{|G|}\sum_{g \in G} g^\top\boldsymbol{\gamma}^*\right\rangle = y_i\langle \overline{\mathbf{x}}_i, \boldsymbol{\gamma}^*\rangle \geq 1. \tag{85}$$

Thus $\frac{1}{|G|}\sum_{g \in G} g^\top\boldsymbol{\gamma}^*$ satisfies the constraints in (65). Assume for the sake of contradiction that $\frac{1}{|G|}\sum_{g \in G} g^\top\boldsymbol{\gamma}^* \neq \boldsymbol{\beta}^*$, then by the uniqueness of the minimizer of (65) we have

$$\|\widetilde{\boldsymbol{\gamma}}\| := \|\text{Proj}_{\text{Range}(\mathcal{A})}\boldsymbol{\beta}^*\| < \left\|\text{Proj}_{\text{Range}(\mathcal{A})}\frac{1}{|G|}\sum_{g \in G} g^\top\boldsymbol{\gamma}^*\right\|, \tag{86}$$

where $\widetilde{\boldsymbol{\gamma}} := \text{Proj}_{\text{Range}(\mathcal{A})}\boldsymbol{\beta}^*$. We make the following two claims to be proved shortly:

- **Claim 1:** $y_i\langle \widetilde{\boldsymbol{\gamma}}, \overline{\mathbf{x}}_i\rangle = y_i\langle \boldsymbol{\beta}^*, \mathbf{x}_i\rangle \geq 1, \forall i \in [n]$.

- **Claim 2:** $\text{Proj}_{\text{Range}(\mathcal{A})} \frac{1}{|G|} \sum_{g \in G} g^\top \boldsymbol{\gamma}^* = \boldsymbol{\gamma}^*$.

These two claims combined with (86) imply that $\widetilde{\boldsymbol{\gamma}}$ satisfies the constraint in (64) and has a smaller norm compared to $\boldsymbol{\gamma}^*$, $\|\widetilde{\boldsymbol{\gamma}}\| < \|\boldsymbol{\gamma}^*\|$. This contradicts the fact that $\boldsymbol{\gamma}^*$ is the minimizer of problem (64). Therefore we have $\frac{1}{|G|} \sum_{g \in G} g^\top \boldsymbol{\gamma}^* = \boldsymbol{\beta}^*$.

If we assume in addition that $\rho_0$ is unitary on $\mathcal{X}_0$, then $\mathcal{A} = \mathcal{A}^\top : \boldsymbol{\beta} \mapsto \overline{\boldsymbol{\beta}}$ is an orthogonal projection from $\mathbb{R}^{d_0}$ onto $\mathbb{R}_G^{d_0} = \text{Range}(\mathcal{A})$. Therefore $\text{Proj}_{\text{Range}(\mathcal{A})} \boldsymbol{\beta} = \boldsymbol{\beta}$ for $\boldsymbol{\beta} \in \mathbb{R}_G^{d_0}$, and hence problems (65) and (66) are equivalent.

Finally, we prove the above two claims.

**Proof of Claim 1:** $y_i \langle \widetilde{\boldsymbol{\gamma}}, \overline{\mathbf{x}}_i \rangle = y_i \langle \boldsymbol{\beta}^*, \mathbf{x}_i \rangle \geq 1, \forall i \in [n]$.

Since $\widetilde{\boldsymbol{\gamma}} = \text{Proj}_{\text{Range}(\mathcal{A})} \boldsymbol{\beta}^*$, we have

$$\widetilde{\boldsymbol{\gamma}} - \boldsymbol{\beta}^* \in \text{Range}(\mathcal{A})^\perp = \ker(\mathcal{A}^\top). \tag{87}$$

Hence

$$\mathcal{A}^\top \widetilde{\boldsymbol{\gamma}} = \mathcal{A}^\top \boldsymbol{\beta}^* = \boldsymbol{\beta}^*, \tag{88}$$

where the second equality is due to $\boldsymbol{\beta}^* \in \mathbb{R}_G^{d_0} = \text{Range}(\mathcal{A}^\top)$ and $\mathcal{A}^\top = \mathcal{A}^\top \circ \mathcal{A}^\top$ is a projection; cf. Proposition 3.6. Therefore, for all $i \in [n]$,

$$y_i \langle \widetilde{\boldsymbol{\gamma}}, \overline{\mathbf{x}}_i \rangle = y_i \langle \mathcal{A}^\top \widetilde{\boldsymbol{\gamma}}, \mathbf{x}_i \rangle = y_i \langle \boldsymbol{\beta}^*, \mathbf{x}_i \rangle \geq 1. \tag{89}$$

**Proof of Claim 2:** $\text{Proj}_{\text{Range}(\mathcal{A})} \mathcal{A}^\top \boldsymbol{\gamma}^* = \boldsymbol{\gamma}^*$.

We first note that both $\text{Proj}_{\text{Range}(\mathcal{A})} \mathcal{A}^\top \boldsymbol{\gamma}^*$ and $\boldsymbol{\gamma}^*$ are in the affine space $\mathcal{A}^\top \boldsymbol{\gamma}^* + \ker(\mathcal{A}^\top)$. Indeed,

$$\left( \mathcal{A}^\top \boldsymbol{\gamma}^* - \text{Proj}_{\text{Range}(\mathcal{A})} \mathcal{A}^\top \boldsymbol{\gamma}^* \right) \in \text{Range}(\mathcal{A})^\perp = \ker(\mathcal{A}^\top), \tag{90}$$

$$\mathcal{A}^\top (\boldsymbol{\gamma}^* - \mathcal{A}^\top \boldsymbol{\gamma}^*) = \mathcal{A}^\top \boldsymbol{\gamma}^* - \mathcal{A}^\top \mathcal{A}^\top \boldsymbol{\gamma}^* = 0. \tag{91}$$

This also implies

$$\mathcal{A}^\top \left( \text{Proj}_{\text{Range}(\mathcal{A})} \mathcal{A}^\top \boldsymbol{\gamma}^* \right) = \mathcal{A}^\top \boldsymbol{\gamma}^*. \tag{92}$$

Moreover, for any $i \in [n]$,

$$y_i \left\langle \text{Proj}_{\text{Range}(\mathcal{A})} \mathcal{A}^\top \boldsymbol{\gamma}^*, \overline{\mathbf{x}}_i \right\rangle = y_i \left\langle \mathcal{A}^\top \left( \text{Proj}_{\text{Range}(\mathcal{A})} \mathcal{A}^\top \boldsymbol{\gamma}^* \right), \mathbf{x}_i \right\rangle \tag{93}$$

$$= y_i \left\langle \mathcal{A}^\top \boldsymbol{\gamma}^*, \mathbf{x}_i \right\rangle = y_i \left\langle \boldsymbol{\gamma}^*, \overline{\mathbf{x}}_i \right\rangle \geq 1. \tag{94}$$

Hence $\text{Proj}_{\text{Range}(\mathcal{A})} \mathcal{A}^\top \boldsymbol{\gamma}^*$ also satisfies the constraint in (64). Since the orthogonal projection $\text{Proj}_{\text{Range}(\mathcal{A})} \mathcal{A}^\top \boldsymbol{\gamma}^*$ achieves the minimal norm among all vectors in the affine space $\mathcal{A}^\top \boldsymbol{\gamma}^* + \ker(\mathcal{A}^\top)$ which includes $\boldsymbol{\gamma}^*$, this can only happen when $\text{Proj}_{\text{Range}(\mathcal{A})} \mathcal{A}^\top \boldsymbol{\gamma}^* = \boldsymbol{\gamma}^*$ is the unique minimizer of (64).

This concludes the proof of Theorem 4.1. □

# D  Proofs in Section 5

**Corollary 5.1.** *Let* $\boldsymbol{\beta}_{\text{steer}}^\infty = \lim_{t \to \infty} \frac{\boldsymbol{\beta}_{\text{steer}}(t)}{\|\boldsymbol{\beta}_{\text{steer}}(t)\|}$ *be the directional limit of the linear predictor* $\boldsymbol{\beta}_{\text{steer}}(t) = \mathcal{P}_{\text{steer}}(\mathbf{W}(t))$ *parameterized by a linear steerable network trained using gradient flow on the **original** data set* $S = \{(\mathbf{x}_i, y_i), i \in [n]\}$. *Correspondingly, let* $\boldsymbol{\beta}_{\text{fc}}^\infty = \lim_{t \to \infty} \frac{\boldsymbol{\beta}_{\text{fc}}(t)}{\|\boldsymbol{\beta}_{\text{fc}}(t)\|}$, $\boldsymbol{\beta}_{\text{fc}}(t) = \mathcal{P}_{\text{fc}}(\mathbf{W}(t))$ *(13), be that of a linear fully-connected network trained on the **augmented** data set* $S_{\text{aug}} = \{(g\mathbf{x}_i, y_i), i \in [n], g \in G\}$. *If* $G$ *acts unitarily on* $\mathcal{X}_0$, *then*

$$\boldsymbol{\beta}_{\text{steer}}^\infty = \boldsymbol{\beta}_{\text{fc}}^\infty. \tag{95}$$

*In other words, the effect of using a linear steerable network for group-invariant binary classification is exactly the same as conducting **data-augmentation** for non-invariant models.*

*Proof.* By Theorem 4.1, a positive scaling $\gamma^* = \tau \beta_{\text{steer}}^\infty$ of $\beta_{\text{steer}}^\infty$ satisfies the KKT condition of (22). That is, there exists $\alpha_i \geq 0$, $i \in [n]$, such that

- *Primal feasibility:*

$$y_i \langle \gamma^*, \overline{\mathbf{x}}_i \rangle \geq 1, \quad \forall i \in [n]. \tag{96}$$

- *Stationarity:*

$$\gamma^* = \sum_{i=1}^n \alpha_i y_i \overline{\mathbf{x}}_i \tag{97}$$

- *Complementary slackness*:

$$y_i \langle \gamma^*, \overline{\mathbf{x}}_i \rangle > 1 \implies \alpha_i = 0. \tag{98}$$

Using Corollary 4.3, we only need to show that $\gamma^*$ is also the solution of

$$\arg \min_{\gamma \in \mathbb{R}^{d_0}} \|\gamma\|^2, \quad \text{s.t. } y_i \langle g\mathbf{x}_i, \gamma \rangle \geq 1, \forall i \in [n], \forall g \in G. \tag{99}$$

That is, there exists $\tilde{\alpha}_{i,g} \geq 0, \forall i \in [n], \forall g \in G$, such that

- *Primal feasibility:*

$$y_i \langle \gamma^*, g\mathbf{x}_i \rangle \geq 1, \quad \forall i \in [n], \forall g \in G. \tag{100}$$

- *Stationarity:*

$$\gamma^* = \sum_{i=1}^n \sum_{g \in G} \tilde{\alpha}_{i,g} y_i g\mathbf{x}_i \tag{101}$$

- *Complementary slackness*:

$$y_i \langle \gamma^*, g\mathbf{x}_i \rangle > 1 \implies \tilde{\alpha}_{i,g} = 0. \tag{102}$$

Indeed, we set $\tilde{\alpha}_{i,g} = \frac{1}{|G|} \alpha_i \geq 0$. Since $\gamma^* \in \mathbb{R}_G^{d_0}$ if $\rho_0$ is unitary (this can also be observed from Eq. (97)), we have $y_i \langle \gamma^*, g\mathbf{x}_i \rangle = y_i \langle \gamma^*, \mathbf{x}_i \rangle, \forall g \in G, \forall i \in [n]$. Hence we have primal feasibility (100) from (96),

$$y_i \langle \gamma^*, g\mathbf{x}_i \rangle = y_i \left\langle \gamma^*, \frac{1}{|G|} \sum_{h \in G} hg\mathbf{x}_i \right\rangle = y_i \langle \gamma^*, \overline{\mathbf{x}}_i \rangle \geq 1, \quad \forall i \in [n], \forall g \in G. \tag{103}$$

Stationarity (101) holds since

$$\sum_{i=1}^n \sum_{g \in G} \tilde{\alpha}_{i,g} y_i g\mathbf{x}_i = \sum_{i=1}^n \sum_{g \in G} \frac{1}{|G|} \alpha_i y_i g\mathbf{x}_i = \sum_{i=1}^n \alpha_i y_i \overline{\mathbf{x}}_i = \gamma^*, \tag{104}$$

where the last equality comes from (97). Finally, if $y_i \langle \gamma^*, g\mathbf{x}_i \rangle > 1$, then (103) and (98) imply that

$$\tilde{\alpha}_{i,g} = \frac{1}{|G|} \alpha_i = 0. \tag{105}$$

This proves the condition for complementary slackness (102). $\qquad\square$

**Corollary 5.5.** *Let $\beta_{\text{steer}}^\infty$ be the same as that in Corollary 5.1. Let $\beta_{\text{fc}}^{\rho_0,\infty} = \lim_{t \to \infty} \frac{\beta_{\text{fc}}^{\rho_0}(t)}{\|\beta_{\text{fc}}^{\rho_0}(t)\|}$ be the limit direction of $\beta_{\text{fc}}^{\rho_0}(t) = \mathcal{P}_{\text{fc}}(\mathbf{W}(t))$ under the gradient flow of the modified empirical loss $\mathcal{L}_{\mathcal{P}_{\text{fc}}}^{\rho_0}(\mathbf{W}; S_{\text{aug}})$ (34) for a linear fully-connected network on the **augmented** data set $S_{\text{aug}}$. Then*

$$\beta_{\text{steer}}^\infty \propto \left( \frac{1}{|G|} \sum_{g \in G} \rho_0(g)^\top \rho_0(g) \right)^{1/2} \beta_{\text{fc}}^{\rho_0,\infty}. \tag{106}$$

*Consequently, we have $\langle \mathbf{x}, \beta_{\text{steer}}^\infty \rangle \propto \langle \mathbf{x}, \beta_{\text{fc}}^{\rho_0,\infty} \rangle_{\rho_0}$ for all $\mathbf{x} \in \mathbb{R}^{d_0}$.*

*Proof.* The proof is similar to that of Corollary 5.1. By Theorem 4.1, the limit direction $\boldsymbol{\beta}^{\infty}_{\text{steer}}$ is proportional to $\frac{1}{|G|} \sum_{g \in G} g^{\top} \boldsymbol{\gamma}^{*}$, where $\boldsymbol{\gamma}^{*}$ satisfies the same KKT condition ((96), (97) and (98)) of problem (22) as in the proof of Corollary 5.1. On the other hand, note that the loss function $\mathcal{L}^{\rho_0}_{\mathcal{P}_{\text{fc}}}(\mathbf{W}; S_{\text{aug}})$ (34) for the modified fully-connected network on the augmented data set $S_{\text{aug}}$ is equivalent to

$$\mathcal{L}^{\rho_0}_{\mathcal{P}_{\text{fc}}}(\mathbf{W}; S_{\text{aug}}) = \sum_{g \in G} \sum_{i=1}^{n} \ell_{\exp} \left( \langle g\mathbf{x}_i, \mathcal{P}_{\text{fc}}(\mathbf{W}) \rangle_{\rho_0}, y_i \right) \tag{107}$$

$$= \sum_{g \in G} \sum_{i=1}^{n} \ell_{\exp} \left( \langle Ag\mathbf{x}_i, \mathcal{P}_{\text{fc}}(\mathbf{W}) \rangle, y_i \right) \tag{108}$$

$$= \mathcal{L}_{\mathcal{P}_{\text{fc}}}(\mathbf{W}; \widetilde{S}_{\text{aug}}), \tag{109}$$

where

$$A = \left( \frac{1}{|G|} \sum_{g \in G} \rho_0(g)^{\top} \rho_0(g) \right)^{1/2}, \quad \widetilde{S}_{\text{aug}} = \{ (Ag\mathbf{x}_i, y_i) : i \in [n], g \in G \}. \tag{110}$$

Therefore, by Corollary 4.3, $\boldsymbol{\beta}^{\rho_0, \infty}_{\text{fc}}$ is proportional to the solution $\widetilde{\boldsymbol{\gamma}}^{*}_{\text{aug}}$ of

$$\widetilde{\boldsymbol{\gamma}}^{*}_{\text{aug}} = \arg \min_{\boldsymbol{\gamma} \in \mathbb{R}^{d_0}} \|\boldsymbol{\gamma}\|^2, \quad \text{s.t. } y_i \langle Ag\mathbf{x}_i, \boldsymbol{\gamma} \rangle \geq 1, \forall i \in [n], \forall g \in G. \tag{111}$$

We claim that $\widetilde{\boldsymbol{\gamma}}^{*}_{\text{aug}} = A^{-1} \left( \frac{1}{|G|} \sum_{g \in G} g^{\top} \boldsymbol{\gamma}^{*} \right)$. To see this, let $\widetilde{\alpha}_{i,g} = \alpha_i, \forall i \in [n]$, where $\alpha_i \geq 0$ is the dual variable for $\boldsymbol{\gamma}^{*}$ of problem (22). We verify below that the KKT conditions for problem (111) are satisfied for the primal-dual pair $A^{-1} \left( \frac{1}{|G|} \sum_{g \in G} g^{\top} \boldsymbol{\gamma}^{*} \right)$ and $(\widetilde{\alpha}_{i,g})_{i \in [n], g \in G}$.

- *Primal feasibility:* for any $i \in [n]$ and $h \in G$,

$$y_i \left\langle A^{-1} \left( \frac{1}{|G|} \sum_{g \in G} g^{\top} \boldsymbol{\gamma}^{*} \right), Ah\mathbf{x}_i \right\rangle = y_i \left\langle \frac{1}{|G|} \sum_{g \in G} g^{\top} \boldsymbol{\gamma}^{*}, h\mathbf{x}_i \right\rangle \tag{112}$$

$$= y_i \left\langle \boldsymbol{\gamma}^{*}, \left( \frac{1}{|G|} \sum_{g \in G} g \right) h\mathbf{x}_i \right\rangle \tag{113}$$

$$= y_i \langle \boldsymbol{\gamma}^{*}, \overline{\mathbf{x}}_i \rangle \geq 1, \tag{114}$$

where the first equality is due to $A^{\top} = A$ being symmetric, and the last inequality is due to (96).

- *Stationarity:*

$$A^{-1}\left(\frac{1}{|G|}\sum_{g\in G}g^\top\boldsymbol{\gamma}^*\right) = A^{-1}\left(\frac{1}{|G|}\sum_{g\in G}g^\top\right)\sum_{i=1}^n\alpha_i y_i\overline{\mathbf{x}}_i \tag{115}$$

$$= A^{-1}\frac{1}{|G|^2}\left(\sum_{g\in G}g^\top\right)\sum_{i=1}^n\alpha_i y_i\sum_{h\in G}h\mathbf{x}_i \tag{116}$$

$$= A\cdot A^{-2}\frac{1}{|G|^2}\sum_{i=1}^n\alpha_i y_i\sum_{g\in G}\sum_{h\in G}g^\top h\mathbf{x}_i \tag{117}$$

$$= A\cdot A^{-2}\frac{1}{|G|}\sum_{i=1}^n\alpha_i y_i\left(\frac{1}{|G|}\sum_{g\in G}g^\top g\right)\sum_{h\in G}h\mathbf{x}_i \tag{118}$$

$$= \sum_{i=1}^n\frac{\alpha_i}{|G|}y_i\sum_{h\in G}Ah\mathbf{x}_i \tag{119}$$

$$= \sum_{i=1}^n\sum_{g\in G}\widetilde{\alpha}_{i,g}y_i Ag\mathbf{x}_i, \tag{120}$$

where the first equality is due to (97), and the last equality comes from the definition of the dual variable $\widetilde{\alpha}_{i,g} = \frac{1}{|G|}\alpha_i, \forall i\in[n], \forall g\in G$.

- *Complementary slackness:* if $y_i\left\langle A^{-1}\left(\frac{1}{|G|}\sum_{g\in G}g^\top\boldsymbol{\gamma}^*\right), Ag\mathbf{x}_i\right\rangle > 1$ for some $i\in[n]$ and $g\in G$, then (114) and (98) imply that $\widetilde{\alpha}_{i,g} = \frac{1}{|G|}\alpha_i = 0$.

Hence $\widetilde{\boldsymbol{\gamma}}^*_{\text{aug}} = A^{-1}\left(\frac{1}{|G|}\sum_{g\in G}g^\top\boldsymbol{\gamma}^*\right)$ is indeed the solution of (111). Therefore,

$$\boldsymbol{\beta}^\infty_{\text{steer}} \propto \frac{1}{|G|}\sum_{g\in G}g^\top\boldsymbol{\gamma}^* = A\boldsymbol{\gamma}^*_{\text{aug}} \propto A\boldsymbol{\beta}^{\rho_0,\infty}_{\text{fc}}. \tag{121}$$

This completes the proof. $\qquad\square$

## E  Proofs of Theorem 6.1

**Theorem 6.1.** *Let $\boldsymbol{\beta}^\infty_{\text{steer}}$ be the directional limit of a linear-steerable-network-parameterized predictor trained on the **original** data set $S = \{(\mathbf{x}_i, y_i), i\in[n]\}$; let $\boldsymbol{\beta}^\infty_{\text{fc}}$ be that of a linear fully-connected network **also** trained on the same data set $S$. Let $M_{\text{steer}}$ and $M_{\text{fc}}$, respectively, be the (signed) margin of $\boldsymbol{\beta}^\infty_{\text{steer}}$ and $\boldsymbol{\beta}^\infty_{\text{fc}}$ on the **augmented** data set $S_{\text{aug}} = \{(g\mathbf{x}_i, y_i) : i\in[n], g\in G\}$, i.e.,*

$$M_{\text{steer}} := \min_{i\in[n], g\in G} y_i\left\langle\boldsymbol{\beta}^\infty_{\text{steer}}, g\mathbf{x}_i\right\rangle, \quad M_{\text{fc}} := \min_{i\in[n], g\in G} y_i\left\langle\boldsymbol{\beta}^\infty_{\text{fc}}, g\mathbf{x}_i\right\rangle. \tag{122}$$

*Then we always have $M_{\text{steer}} \geq M_{\text{fc}}$.*

*Proof.* By Corollary 4.3 and Corollary 5.1, we have

$$\boldsymbol{\beta}^\infty_{\text{steer}} \propto \boldsymbol{\gamma}^*_{\text{steer}} = \arg\min_{\boldsymbol{\gamma}\in\mathbb{R}^{d_0}}\|\boldsymbol{\gamma}\|^2, \quad \text{s.t. } y_i\left\langle\boldsymbol{\gamma}, g\mathbf{x}_i\right\rangle \geq 1, \forall g\in G, \forall i\in[n], \tag{123}$$

$$\boldsymbol{\beta}^\infty_{\text{fc}} \propto \boldsymbol{\gamma}^*_{\text{fc}} = \arg\min_{\boldsymbol{\gamma}\in\mathbb{R}^{d_0}}\|\boldsymbol{\gamma}\|^2, \quad \text{s.t. } y_i\left\langle\boldsymbol{\gamma}, \mathbf{x}_i\right\rangle \geq 1, \forall i\in[n]. \tag{124}$$

Moreover, the margin $M_{\text{steer}}$ of the steerable network $\boldsymbol{\beta}^\infty_{\text{steer}}$ on the augmented data set $S_{\text{aug}}$ is $M_{\text{steer}} = \frac{1}{\|\boldsymbol{\gamma}^*_{\text{steer}}\|}$. Consider the following three cases.

- *Case 1:* $\min_{i\in[n], g\in G} y_i\left\langle\boldsymbol{\gamma}^*_{\text{fc}}, g\mathbf{x}_i\right\rangle \geq 1$. In this case, $\boldsymbol{\gamma}^*_{\text{fc}}$ also satisfies the more restrictive constraint in (123). Therefore $\boldsymbol{\gamma}^*_{\text{fc}} = \boldsymbol{\gamma}^*_{\text{steer}}$, which implies

$$\boldsymbol{\beta}^\infty_{\text{steer}} = \boldsymbol{\beta}^\infty_{\text{fc}}, \quad \text{and} \quad M_{\text{steer}} = M_{\text{fc}}. \tag{125}$$

- *Case 2:* $\min_{i\in[n], g\in G} y_i \langle \boldsymbol{\gamma}^*_{\text{fc}}, g\mathbf{x}_i \rangle \leq 0$. This implies that $\boldsymbol{\gamma}^*_{\text{fc}}$, and hence $\boldsymbol{\beta}^\infty_{\text{fc}}$ has a non-positive margin on $D_{\text{aug}}$. Thus

$$M_{\text{fc}} \leq 0 < M_{\text{steer}}. \tag{126}$$

- *Case 3:* $0 < \min_{i\in[n], g\in G} y_i \langle \boldsymbol{\gamma}^*_{\text{fc}}, g\mathbf{x}_i \rangle < 1$. In this case, define

$$\widetilde{\boldsymbol{\gamma}}^*_{\text{fc}} := \frac{\boldsymbol{\gamma}^*_{\text{fc}}}{\min_{i\in[n], g\in G} y_i \langle \boldsymbol{\gamma}^*_{\text{fc}}, g\mathbf{x}_i \rangle}. \tag{127}$$

Then $\widetilde{\boldsymbol{\gamma}}^*_{\text{fc}}$ satisfies the constraint in (123), and hence

$$\|\widetilde{\boldsymbol{\gamma}}^*_{\text{fc}}\| \geq \|\boldsymbol{\gamma}^*_{\text{steer}}\|. \tag{128}$$

Therefore

$$M_{\text{fc}} = \min_{i\in[n], g\in G} y_i \langle \boldsymbol{\beta}^\infty_{\text{fc}}, g\mathbf{x}_i \rangle = \frac{\min_{i\in[n], g\in G} y_i \langle \boldsymbol{\gamma}^*_{\text{fc}}, g\mathbf{x}_i \rangle}{\|\boldsymbol{\gamma}^*_{\text{fc}}\|} = \frac{1}{\|\widetilde{\boldsymbol{\gamma}}^*_{\text{fc}}\|} \tag{129}$$

$$\leq \frac{1}{\|\boldsymbol{\gamma}^*_{\text{steer}}\|} = M_{\text{steer}}. \tag{130}$$

This completes the proof. $\qquad\square$

## F  Proof of Theorem 6.3

To prove Theorem 6.3, we need first the following preliminaries.

**Lemma F.1.** *If $\mathcal{D}$ is $G$-invariant and linearly separable, then $\mathcal{D}$ can be linearly separated by a $G$-invariant classifier. That is, there exists $\boldsymbol{\beta} \in \mathbb{R}^{d_0}_G$ such that*

$$\mathbb{P}_{(\mathbf{x},y)\sim\mathcal{D}} [y \langle \mathbf{x}, \boldsymbol{\beta} \rangle \geq 1] = 1. \tag{131}$$

*Proof.* Since $\mathcal{D}$ is linearly separable, there exists $\boldsymbol{\beta}_0 \in \mathbb{R}^{d_0}$ such that $\mathbb{P}_{(\mathbf{x},y)\sim\mathcal{D}} [y \langle \mathbf{x}, \boldsymbol{\beta}_0 \rangle \geq 1] = 1$. Let $\boldsymbol{\beta} = \frac{1}{|G|} \sum_{g\in G} g^\top \boldsymbol{\beta}_0 \in \mathbb{R}^{d_0}_G$, and we aim to show that $\mathcal{D}$ can also be linearly separated by $\boldsymbol{\beta}$. Indeed, for all $(\mathbf{x}, y) \in \mathbb{R}^{d_0} \times \{\pm 1\}$,

$$y \langle \mathbf{x}, \boldsymbol{\beta} \rangle < 1 \implies y \left\langle \mathbf{x}, \frac{1}{|G|} \sum_{g\in G} g^\top \boldsymbol{\beta}_0 \right\rangle < 1 \implies y \left\langle \frac{1}{|G|} \sum_{g\in G} g\mathbf{x}, \boldsymbol{\beta}_0 \right\rangle < 1 \tag{132}$$

$$\implies \exists g \in G, \text{ s.t. } y \langle g\mathbf{x}, \boldsymbol{\beta}_0 \rangle < 1. \tag{133}$$

Let $E = \{(\mathbf{x}, y) : y \langle \mathbf{x}, \boldsymbol{\beta}_0 \rangle < 1\} \subset \mathbb{R}^{d_0} \times \{\pm 1\}$, then

$$\mathbb{P}_{(\mathbf{x},y)\sim\mathcal{D}} [y \langle \mathbf{x}, \boldsymbol{\beta} \rangle < 1] \leq \sum_{g\in G} \mathbb{P}_{(\mathbf{x},y)\sim\mathcal{D}} [y \langle g\mathbf{x}, \boldsymbol{\beta}_0 \rangle < 1] \tag{134}$$

$$= \sum_{g\in G} \mathcal{D} \{(\mathbf{x}, y) : (\rho_0(g) \otimes \text{Id})(\mathbf{x}, y) \in E\} \tag{135}$$

$$= \sum_{g\in G} (\rho_0(g) \otimes \text{Id})_* \mathcal{D}(E) = \sum_{g\in G} \mathcal{D}(E) \tag{136}$$

$$= \sum_{g\in G} \mathbb{P}_{(\mathbf{x},y)\sim\mathcal{D}} [y \langle \mathbf{x}, \boldsymbol{\beta}_0 \rangle < 1] = 0, \tag{137}$$

where the second equality in (136) is due to $\mathcal{D}$ being $G$-invariant. Therefore $\mathcal{D}$ can be separated by $\boldsymbol{\beta} \in \mathbb{R}^{d_0}_G$. $\qquad\square$

**Definition F.2** (Empirical Rademacher complexity). *Let $\mathcal{F} \subset \mathbb{R}^{\mathcal{Z}}$ be a class of real-valued functions on $\mathcal{Z}$, and let $S = (\mathbf{z}_1, \cdots, \mathbf{z}_n)$ be a set of $n$ samples from $\mathcal{Z}$. The (empirical) Rademacher complexity $\widehat{\mathcal{R}}_n(\mathcal{F}, S)$ of $\mathcal{F}$ over $S$ is defined as*

$$\widehat{\mathcal{R}}_n(\mathcal{F}, S) := \frac{1}{n} \mathbb{E}_{\boldsymbol{\sigma}\sim\{\pm 1\}^n} \left[ \sup_{f\in\mathcal{F}} \sum_{i=1}^n \sigma_i f(\mathbf{z}_i) \right], \tag{138}$$

*where $\boldsymbol{\sigma} = (\sigma_1, \cdots, \sigma_n)$ are the i.i.d. Rademacher variables satisfying $\mathbb{P}[\sigma_i = 1] = \mathbb{P}[\sigma_i = -1] = 1/2$.*

**Lemma F.3** (Talagrand's Contraction Lemma, Lemma 8 in Mohri and Medina [2014]). *Let* $\Phi_i :$ $\mathbb{R} \to \mathbb{R}$, $i \in [n]$, *be 1-Lipschitz functions,* $\mathcal{F} \subset \mathbb{R}^{\mathcal{Z}}$ *be a function class, and* $S = (\mathbf{z}_1, \ldots, \mathbf{z}_n) \in \mathcal{Z}^n$ *be a set of $n$ samples from $\mathcal{Z}$. Then*

$$\widehat{\mathcal{R}}_n(\mathcal{F}, S) \geq \frac{1}{n} \mathbb{E}_{\boldsymbol{\sigma} \sim \{\pm 1\}^n} \left[ \sup_{f \in \mathcal{F}} \sum_{i=1}^{n} \sigma_i \Phi_i \circ f(\mathbf{z}_i) \right]. \tag{139}$$

**Lemma F.4.** *Let* $\mathcal{H} = \left\{ \mathbf{x} \mapsto \langle \boldsymbol{\beta}, \mathbf{x} \rangle : \boldsymbol{\beta} \in \mathbb{R}_G^{d_0}, \|\boldsymbol{\beta}\| \leq B \right\}$ *be the function space of $G$-invariant linear functions of bounded norm, and let* $S_{\mathbf{x}} = \{\mathbf{x}_1, \cdots, \mathbf{x}_n\} \subset \mathbb{R}^{d_0}$ *be a set of $n$ samples from* $\mathbb{R}^{d_0}$. *Then*

$$\widehat{\mathcal{R}}_n(\mathcal{H}, S_{\mathbf{x}}) \leq \frac{B \max_{i \in [n]} \|\overline{\mathbf{x}}_i\|}{\sqrt{n}} \tag{140}$$

*Proof.* By the definition of empirical Rademacher complexity (138), we have

$$n\widehat{\mathcal{R}}_n(\mathcal{H}, S_{\mathbf{x}}) = \mathbb{E}_{\boldsymbol{\sigma}} \left[ \sup_{\boldsymbol{\beta} \in \mathbb{R}_G^{d_0}, \|\boldsymbol{\beta}\| \leq B} \sum_{i=1}^{n} \sigma_i \langle \boldsymbol{\beta}, \mathbf{x}_i \rangle \right] \tag{141}$$

$$= \mathbb{E}_{\boldsymbol{\sigma}} \left[ \sup_{\boldsymbol{\beta} \in \mathbb{R}_G^{d_0}, \|\boldsymbol{\beta}\| \leq B} \sum_{i=1}^{n} \sigma_i \langle \boldsymbol{\beta}, \overline{\mathbf{x}}_i \rangle \right] \tag{142}$$

$$= \mathbb{E}_{\boldsymbol{\sigma}} \sup_{\boldsymbol{\beta} \in \mathbb{R}_G^{d_0}, \|\boldsymbol{\beta}\| \leq B} \left\langle \boldsymbol{\beta}, \sum_{i=1}^{n} \sigma_i \overline{\mathbf{x}}_i \right\rangle \tag{143}$$

$$= B\mathbb{E}_{\boldsymbol{\sigma}} \left\| \sum_{i=1}^{n} \sigma_i \overline{\mathbf{x}}_i \right\| \tag{144}$$

$$\leq B \left( \mathbb{E}_{\boldsymbol{\sigma}} \left\| \sum_{i=1}^{n} \sigma_i \overline{\mathbf{x}}_i \right\|^2 \right)^{1/2}, \tag{145}$$

where (142) is due to $\boldsymbol{\beta} \in \mathbb{R}_G^{d_0}$. Since $\sigma_1, \cdots, \sigma_n$ are i.i.d., we have

$$\mathbb{E}_{\boldsymbol{\sigma}} \left\| \sum_{i=1}^{n} \sigma_i \overline{\mathbf{x}}_i \right\|^2 = \mathbb{E}_{\boldsymbol{\sigma}} \left[ \sum_{i,j} \sigma_i \sigma_j \langle \overline{\mathbf{x}}_i, \overline{\mathbf{x}}_j \rangle \right] \tag{146}$$

$$= \sum_{i \neq j} \langle \overline{\mathbf{x}}_i, \overline{\mathbf{x}}_j \rangle \mathbb{E}_{\boldsymbol{\sigma}} \sigma_i \sigma_j + \sum_{i=1}^{n} \|\overline{\mathbf{x}}_i\|^2 \mathbb{E}_{\boldsymbol{\sigma}} \sigma_i^2 \tag{147}$$

$$= \sum_{i=1}^{n} \|\overline{\mathbf{x}}_i\|^2 \leq n \max_{i \in [n]} \|\overline{\mathbf{x}}_i\|^2. \tag{148}$$

Eq. (145) combined with (148) completes the proof. $\square$

We also need the following standard result on the generalization bound based on the Rademacher complexity [Shalev-Shwartz and Ben-David, 2014].

**Lemma F.5.** *Let $\mathcal{H}$ be a set of hypotheses,* $\mathcal{Z} = \mathbb{R}^{d_0} \times \{\pm 1\}$ *be the set of labeled samples, and*

$$\ell : \mathcal{H} \times \mathcal{Z} \to [0, \infty), \quad (h, \mathbf{z}) \mapsto \ell(h, \mathbf{z}), \tag{149}$$

*be a loss function satisfying $0 \leq \ell(h, \mathbf{z}) \leq c$ for all $\mathbf{z} \in \mathcal{Z}$ and $h \in \mathcal{H}$. Define the function class* $\mathcal{F} = \ell(\mathcal{H}, \cdot) := \{\mathbf{z} \mapsto \ell(h, \mathbf{z}) : h \in \mathcal{H}\}$. *Then, for any $\delta > 0$ and any distribution $\mathcal{D}$ on $\mathcal{Z}$, we have with probability at least $1 - \delta$ over i.i.d. samples $S = (\mathbf{z}_1, \cdots, \mathbf{z}_n) \sim \mathcal{D}^n$ that*

$$\sup_{h \in \mathcal{H}} \mathcal{L}_{\mathcal{D}}(h) - \mathcal{L}_S(h) \leq 2\mathcal{R}_n(\mathcal{F}) + c\sqrt{\frac{\log(1/\delta)}{2n}}, \tag{150}$$

where $\mathcal{L}_{\mathcal{D}}(h) := \mathbb{E}_{\mathbf{z}\sim\mathcal{D}}\ell(h, \mathbf{z})$ and $\mathcal{L}_S(h) := \frac{1}{n}\sum_{i=1}^n \ell(h, \mathbf{z}_i)$ are, respectively, the population and empirical loss of a hypothesis $h \in \mathcal{H}$, and $\mathcal{R}_n(\mathcal{F}) := \mathbb{E}_{S\sim\mathcal{D}^n}\widehat{\mathcal{R}}_n(\mathcal{F}, S)$ is the Rademacher complexity of $\mathcal{F}$.

Finally, we can prove Theorem 6.3 restated below

**Theorem 6.3.** *Let $\mathcal{D}$ be a $G$-invariant distribution over $\mathbb{R}^{d_0} \times \{\pm 1\}$ that is linearly separable by an invariant classifier $\boldsymbol{\beta}_0 \in \mathbb{R}_G^{d_0}$. Define*

$$\overline{R} = \inf\{r > 0 : \|\overline{\mathbf{x}}\| \leq r \text{ with probability } 1\}. \tag{151}$$

*Let $S = \{(\mathbf{x}_i, y_i)\}_{i=1}^n$ be i.i.d. samples from $\mathcal{D}$, and let $\boldsymbol{\beta}_{\text{steer}}^\infty$ be the limit direction of a steerable-network-parameterized linear predictor trained using gradient flow on $S$. Then, for any $\delta > 0$, we have with probability at least $1 - \delta$ (over random samples $S \sim \mathcal{D}^n$) that*

$$\mathbb{P}_{(\mathbf{x},y)\sim\mathcal{D}}\left[y \neq \text{sign}\left(\langle\mathbf{x}, \boldsymbol{\beta}_{\text{steer}}^\infty\rangle\right)\right] \leq \frac{2\overline{R}\|\boldsymbol{\beta}_0\|}{\sqrt{n}} + \sqrt{\frac{\log(1/\delta)}{2n}}. \tag{152}$$

*Proof of Theorem 6.3.* Let $\mathcal{H} = \{\boldsymbol{\beta} \in \mathbb{R}_G^{d_0} : \|\boldsymbol{\beta}\| \leq \|\boldsymbol{\beta}_0\|\}$ and with slight abuse of notation we identify $\boldsymbol{\beta} \in \mathcal{H}$ with the invariant linear map $\mathbf{x} \mapsto \langle\mathbf{x}, \boldsymbol{\beta}\rangle$. According to Theorem 4.1, let $\boldsymbol{\beta}^* = \tau\boldsymbol{\beta}_{\text{steer}}^\infty$, $\tau > 0$, be the minimizer of (24). Since by assumption $\boldsymbol{\beta}_0$ also satisfies the constraint in (24), we have $\|\boldsymbol{\beta}^*\| \leq \|\boldsymbol{\beta}_0\|$, and hence $\boldsymbol{\beta}^* \in \mathcal{H}$.

Consider the *ramp* loss $\ell$ defined as

$$\ell : \mathcal{H} \times (\mathbb{R}^{d_0} \times \{\pm 1\}) \to [0, 1], \quad (\boldsymbol{\beta}, (\mathbf{x}, y)) \mapsto \min\{1, \max\{0, 1 - y\langle\mathbf{x}, \boldsymbol{\beta}\rangle\}\}. \tag{153}$$

It is easy to verify that $\ell$ upper bounds the 0-1 loss: for all $\boldsymbol{\beta} \in \mathcal{H}$ and $(\mathbf{x}, y) \in \mathbb{R}^{d_0} \times \{\pm 1\}$,

$$\ell(\boldsymbol{\beta}, (\mathbf{x}, y)) \geq \mathbb{1}_{\{y \neq \text{sign}(\langle\mathbf{x},\boldsymbol{\beta}\rangle)\}}. \tag{154}$$

This implies that

$$\mathbb{P}_{(\mathbf{x},y)\sim\mathcal{D}}\left[y \neq \text{sign}\left(\langle\mathbf{x}, \boldsymbol{\beta}_{\text{steer}}^\infty\rangle\right)\right] = \mathbb{P}_{(\mathbf{x},y)\sim\mathcal{D}}\left[y \neq \text{sign}\left(\langle\mathbf{x}, \boldsymbol{\beta}^*\rangle\right)\right] \tag{155}$$

$$= \mathbb{E}_{(\mathbf{x},y)\sim\mathcal{D}}\mathbb{1}_{\{y \neq \text{sign}(\langle\mathbf{x},\boldsymbol{\beta}^*\rangle)\}} \tag{156}$$

$$\leq \mathbb{E}_{(\mathbf{x},y)\sim\mathcal{D}}\ell(\boldsymbol{\beta}^*, (\mathbf{x}, y)) = \mathcal{L}_{\mathcal{D}}(\boldsymbol{\beta}^*). \tag{157}$$

Since $\mathcal{L}_S(\boldsymbol{\beta}^*)$ is always 0 by definition (24) and $\boldsymbol{\beta}^* \in \mathcal{H}$, we have by Lemma F.5 that

$$\mathbb{P}_{(\mathbf{x},y)\sim\mathcal{D}}\left[y \neq \text{sign}\left(\langle\mathbf{x}, \boldsymbol{\beta}_{\text{steer}}^\infty\rangle\right)\right] \leq \mathcal{L}_{\mathcal{D}}(\boldsymbol{\beta}^*) = \mathcal{L}_{\mathcal{D}}(\boldsymbol{\beta}^*) - \mathcal{L}_S(\boldsymbol{\beta}^*) \leq \sup_{\boldsymbol{\beta}\in\mathcal{H}}\mathcal{L}_{\mathcal{D}}(\boldsymbol{\beta}) - \mathcal{L}_S(\boldsymbol{\beta}) \tag{158}$$

$$\leq 2\mathcal{R}_n(\mathcal{F}) + \sqrt{\frac{\log(1/\delta)}{2n}}, \tag{159}$$

where $\mathcal{F} = \ell(\mathcal{H}, \cdot) = \{(\mathbf{x}, y) \mapsto \ell(\boldsymbol{\beta}, (\mathbf{x}, y)) : \boldsymbol{\beta} \in \mathcal{H}\}$. Therefore, to prove Theorem 6.3 it suffices to show that $\mathcal{R}_n(\mathcal{F}) \leq \frac{\overline{R}\|\boldsymbol{\beta}_0\|}{\sqrt{n}}$. To this end, let $S = \{(\mathbf{x}_i, y_i) : i \in [n]\}$ be a set of $n$ labeled samples, and let $S_{\mathbf{x}} = \{\mathbf{x}_i : i \in [n]\}$ be the corresponding set of unlabeled inputs, then

$$\widehat{\mathcal{R}}_n(\mathcal{F}, S) = \frac{1}{n}\mathbb{E}_{\boldsymbol{\sigma}}\sup_{f\in\mathcal{F}}\sum_{i=1}^n \sigma_i f(\mathbf{x}_i, y_i) \tag{160}$$

$$= \frac{1}{n}\mathbb{E}_{\boldsymbol{\sigma}}\sup_{\boldsymbol{\beta}\in\mathcal{H}}\sum_{i=1}^n \sigma_i \ell(\boldsymbol{\beta}, (\mathbf{x}_i, y_i)) \tag{161}$$

$$= \frac{1}{n}\mathbb{E}_{\boldsymbol{\sigma}}\sup_{\boldsymbol{\beta}\in\mathcal{H}}\sum_{i=1}^n \sigma_i \min\{1, \max\{0, 1 - y_i\langle\mathbf{x}_i, \boldsymbol{\beta}\rangle\}\} \tag{162}$$

$$= \frac{1}{n}\mathbb{E}_{\boldsymbol{\sigma}}\sup_{\boldsymbol{\beta}\in\mathcal{H}}\sum_{i=1}^n \sigma_i \Phi_i(\langle\mathbf{x}_i, \boldsymbol{\beta}\rangle), \tag{163}$$

where $\Phi_i : \mathbb{R} \to \mathbb{R}, a \mapsto \min\{1, \max\{0, 1 - y_i a\}\}$, is 1-Lipschitz. Lemma F.3 thus implies that

$$\widehat{\mathcal{R}}_n(\mathcal{F}, S) \leq \widehat{\mathcal{R}}_n(\mathcal{H}, S_{\mathbf{x}}). \tag{164}$$

Using Lemma F.4 and the fact that $\overline{\mathbf{x}} \leq \overline{R}$ with probability 1 over $(\mathbf{x}, y) \sim \mathcal{D}$, we arrive at

$$\mathcal{R}_n(\mathcal{F}) = \mathbb{E}_{S \sim \mathcal{D}^n} \widehat{\mathcal{R}}_n(\mathcal{F}, S) \leq \mathbb{E}_{S \sim \mathcal{D}^n} \widehat{\mathcal{R}}_n(\mathcal{H}, S_{\mathbf{x}}) \leq \frac{\overline{R} \|\boldsymbol{\beta}_0\|}{\sqrt{n}}. \tag{165}$$

This completes the proof of Theorem 6.3. $\qquad\qquad\qquad\qquad\qquad\qquad\qquad\qquad\square$

