# OpenReview forum: "On the Implicit Bias of Linear Equivariant Steerable Networks"
_NeurIPS.cc/2023/Conference — NeurIPS 2023 poster_

### Official Review · Reviewer_H6K3 · 2023-07-05

**Soundness:** 2 fair
**Presentation:** 3 good
**Contribution:** 3 good
**Rating:** 5
**Confidence:** 3

**Summary:**

This paper studies an implicit bias of gradient descent (GD) on linear equivariant neural networks. The authors show that the asymptotic GD solution of a linear steerable neural network, an instance of the equivariant nets, agrees with that of an FC net trained with data augmentation. They also show that the former achieves better margin and generalization bound than the latter.

**Strengths:**

1. This study tackles a critical problem --- analysis of the solution of equivariant neural networks.
2. The paper is well-written and easy to follow.
3. The theory is presented cleanly and solidly. I found a few typos, but The mostly I could read the theory part without confusion.

**Weaknesses:**

1. The paper does not contain empirical evaluations. The authors admit that the analysis relies on the asymptotic approximation, which might have a gap toward a real solution. I feel the experiments make the paper significantly more convincing.
2. The paper limits the analysis to linear cases, but nonlinear networks are practically essential. Extending the theory toward nonlinearity might not be trivial, but at least we can evaluate them through experiments.

**Questions:**

1. The implications of Theorem 4.1 and Corollary 5.1 are interesting. Theorem 4.1 says that the GD solution of a steerable net converges to the max margin solution trained on the group-averaged data $\bar{x}$. In contrast, Corollary 5.1 shows an FC net trained on the group-augmented data converges to the steerable net, which suggests that group averaging in the data space and data augmentation have roughly the same effect. This is not trivial because group averaging takes all possible augmentations for each data instance, but data augmentation puts a single transformation. In the sense of loss function, the former is written as $\mathrm{loss}(E_{g}[gx])$ while the latter is $E_{g}[\mathrm{loss}(gx)]$. Can you explain why these two are possibly equivalent?
2. Related to the above question, I feel that group averaging is somehow similar to mixup, which augments data as $\lambda x + (1-\lambda) x'$ for $\lambda \in [0, 1]$. We may write group averaging as a special case of mixup, where we take an average in terms of groups instead of data instances. Do you have any insight in this direction?
3. In line 186--187 $\beta$ should be in $R^{d_0}_G$ instead of $R^{d_0}$
4. Corollary 5.1: We may need some conditions for gs used in $S_aug$, such as g is drawn uniformly randomly from G.

---

> ### Author Rebuttal · Authors · 2023-08-09
>
> **Comment 1:** The paper does not contain empirical evaluations. The authors admit that the analysis relies on the asymptotic approximation, which might have a gap toward a real solution. I feel the experiments make the paper significantly more convincing.
>
> **Answer:**  We agree with the reviewer that empirical verification of the theory established in this paper could  substantively enhance the impact of our theoretical findings. However, a notable limitation of our findings is that the implicit bias of gradient flow, as studied here, holds true only in an asymptotic sense. The convergence rate to the directional limit is, in fact, exponentially slow. This is consistent with the findings in, Ref [1,2]. Specifically, Ref [2] shows that even if the loss function reaches an order of $10^{-5}$, the limit direction of the parameterized classifier (of a linear fully-connected network) is dependent on the initialization. This is because the networks are not trained all the way to zero loss. Understanding the behavior of gradient flow in a non-asymptotic regime is an important direction for future work.
>
> We have mentioned this limitation in the conclusion of our paper, and we will emphasize this point further in the main text of our revised manuscript.
>
> [1] D. Soudry, E. Hoffer, M. S. Nacson, S. Gunasekar, and N. Srebro. The implicit bias of gradient descent on separable data. The Journal of Machine Learning Research, 19(1):2822–2878, 2018.
>
> [2] C. Yun, S. Krishnan, and H. Mobahi. A unifying view on implicit bias in training linear neural networks. In International Conference on Learning Representations, 2021
>
> **Comment 2:** The paper limits the analysis to linear cases, but nonlinear networks are practically essential. Extending the theory toward nonlinearity might not be trivial, but at least we can evaluate them through experiments.
>
> **Answer:**  The investigation of implicit bias in linear networks typically functions as a foundational step in exploring the more intricate nature of implicit bias in deep nonlinear networks. Despite a wave of recent research efforts to elucidate the implicit bias in nonlinear structures, the field still presents many unresolved questions.
>
> The extension of our current theory to encompass nonlinearity is indeed a vital direction but goes beyond the scope of our present study. Nonetheless, an intriguing theoretical insight that our paper's findings may offer is the possibility that equivariant nonlinear networks could attain a larger margin in the **parameter space** relative to their non-equivariant counterparts. This notion is hinted at in Theorem 6.1 of our work. Future research may further explore this promising direction, building on the groundwork laid in our current analysis.
>
> **Comment 3:** The implications of Theorem 4.1 and Corollary 5.1 are interesting. Theorem 4.1 says that the GD solution of a steerable net converges to the max margin solution trained on the group-averaged data $\bar{\mathbf{x}}$. In contrast, Corollary 5.1 shows an FC net trained on the group-augmented data converges to the steerable net, which suggests that group averaging in the data space and data augmentation have roughly the same effect. This is not trivial because group averaging takes all possible augmentations for each data instance, but data augmentation puts a single transformation. In the sense of loss function, the former is written as $loss(E_g[gx])$ while the latter is $E_g[loss(gx)]$. Can you explain why these two are possibly equivalent?
>
> **Answer:**  We thank the reviewer for posing this insightful question. You are absolutely correct in identifying the two distinct objective functions that correspond to training with data augmentation and training with averaged inputs. While it is true that these two objective functions lead to different training dynamics, our study essentially demonstrates that the directional limits of the parameterized classifiers, when trained under gradient flow, are identical. We do note that this holds true because of our assumption that the networks only have linear layers, and the group actions are also linear.
>
> This insight reveals an interesting connection between these seemingly disparate approaches, and we will make sure to include this observation in the revised version of our paper.
>
> **Comment 4:** Related to the above question, I feel that group averaging is somehow similar to mixup, which augments data as $\lambda x + (1-\lambda)x'$. We may write group averaging as a special case of mixup, where we take an average in terms of groups instead of data instances. Do you have any insight in this direction?
>
> **Answer:** This is an interesting question. After the ``mixup" augmentation, we might have a much larger training set (including pairs of $x$ and $x'$, as well as randomly sampled $\lambda$.) Unfortunately, at this stage, we do not possess direct insight into the ramifications of this augmentation technique. It presents an intriguing avenue for exploration, but elucidating its full impact would require further investigation.
>
> **Comment 5:** Typo.
>
> **Answer:** thank you for spotting that. This will be fixed in the revision.
>
> **Comment 6:** Corollary 5.1: We may need some conditions for $g$s, such as $g$ is drawn uniformly randomly from $G$.
>
> **Answer:** In our paper, we specifically consider the ``full" data augmentation approach that enlarges the dataset by a factor equal to the size of the group $|G|$. In this context, the element $g$ is not randomly drawn, but rather systematically used to encompass the entire group structure. This ensures a comprehensive and deterministic treatment, rather than a stochastic one. We will clarify this aspect further in our revision.

---

> > ### Comment · Reviewer_H6K3 · 2023-08-14
> > **Response**
> >
> > Thank you for the response. I appreciate your candid response. However, my concern about the empirical evaluation remains. If the actual behavior is far different from what we expect from the theory, the impact of this study becomes less significant. I believe some small-scale experiments are necessary to know the truth. Regarding this point, I will keep my score.

---

> > > ### Author Response · Authors · 2023-08-15
> > > **Thanks**
> > >
> > > We thank the reviewer for reading through our response.

---

### Official Review · Reviewer_nbB9 · 2023-07-05

**Soundness:** 4 excellent
**Presentation:** 2 fair
**Contribution:** 3 good
**Rating:** 6
**Confidence:** 2

**Summary:**

The paper present several results on linear equivariant steerable networks. 1) linear equivariant NN converge in the same direction as linear networks trained with data-augmentation, 2) equivariant networks always attain a larger margin then group-augmented fully connected networks, 3) also the generalization bounds are better for equivariant networks when the data distributions is group invariant.

* score has been updated after the response to my assessment.

**Strengths:**

* The paper presents a collection of statements about equivariant linear networks which could be useful as a general reference

* Although the paper is extremely dense in information, it is readable and self-contained. The supplementary materials provide all the proofs.

**Weaknesses:**

1. The paper is essentially a mathematics paper and is the question to what extend it fits the scope of NeurIPS. As a non-mathematician, but someone interested in these topics, I did verify some of the proofs and the overall presentation is clear, however, I find it hard to assess the significance of the work.

2. In particular, the limited scope on linear layers almost make the result seem trivial (though not from the mathematical perspective). To what extend do these results give insights on non-linear, multi-layer equivariant neural networks?

**Questions:**

1. Why is the emphasis on equivariant *steerable* networks? (instead of just equivariant networks) Steerability is I suppose a matter of definition, but classically the difference lies in whether you use the regular representation (G-convs) or irreducible representations (steerable G-convs) to generate the G-feature maps. Since the paper primarily focusses on the regular representation, I do not see why it is important to emphasize steerability. Moreover, line 79 says “more generally, steerable”, but I don’t think steerable G-CNNs are more general per se. That is, any linear layer is equivariant iff it is a group convolution [bekkers2018, thm1] regardless of the used feature type (one can always convert between regular representations and irreps via a generalized Fourier transform). See e.g. WeilerAndCesa 2019 or Brandstetter et al 2022 where this is recently discussed.

Bekkers, E. J. (2019, September). B-Spline CNNs on Lie groups. In International Conference on Learning Representations.

Weiler, M., & Cesa, G. (2019). General e (2)-equivariant steerable cnns. Advances in neural information processing systems, 32.

Brandstetter, J., Hesselink, R., van der Pol, E., Bekkers, E. J., & Welling, M. (2021, October). Geometric and Physical Quantities improve E (3) Equivariant Message Passing. In International Conference on Learning Representations.

2. How does the following reference relate to the current work? It is also about generalization bounds of equivariant networks (through PAC learning) and thus seems related.

Elesedy, B. (2022, March). Group symmetry in PAC learning. In ICLR 2022 Workshop on Geometrical and Topological Representation Learning.

**Limitations:**

Limitations are adequately addressed.

---

> ### Author Rebuttal · Authors · 2023-08-09
>
> **Comment 1:** The paper is essentially a mathematics paper and is the question to what extend it fits the scope of NeurIPS. I find it hard to assess the significance of the work.
>
> **Answer:** Our research is indeed theoretical in nature, and we think it fits within the scope of NeurIPS, as reflected in similar works appearing in top ML conferences; see Ref. [1-4] for instance.
>
> Our work studies the training dynamics and inductive biases of NNs, a key aspect in explaining their generalization. Although there is a vast empirical body of work showcasing the improved generalization of equivariant NNs in group-invariant learning tasks, our paper is unique in rigorously showing that these networks indeed converge to a better solution when compared to their non-equivariant counterparts.
>
> Moreover, we have shown that in the framework we explore, data augmentation is equivalent to learning using equivariant nets. This has long been a postulated hypothesis but, to our knowledge, not rigorously proven until now.
>
> We believe our findings contribute significantly to the field, providing new insights that have been suspected but not previously confirmed.
>
> **Comment 2:** The limited scope on linear layers make the result seem trivial. To what extend do these results give insights on non-linear, multi-layer equivariant NNs?
>
> **Answer:**  The study of implicit bias in linear nets often serves as an initial step towards unraveling the complexities of implicit bias in deep nonlinear nets. While there has been a surge of recent research aiming to explain the implicit bias in nonlinear nets, the field remains largely open. A particular insight gleaned from our paper's results is that equivariant nonlinear nets may achieve a larger margin in the **parameter space** compared to their non-equivariant counterparts, as alluded to in Theorem 6.1.
>
> We respectfully disagree with the reviewer's assessment that the limited focus on linear layers renders our results trivial. On the contrary, as elaborated in response to the previous comment, our theoretical findings offer fresh perspectives on the generalization capabilities of equivariant networks, as well as their relationship to data augmentation. These insights provide tangible advances in understanding concepts that have been speculated upon but not definitively proven until now.
>
> **Comment 3:** Since the paper primarily focusses on the regular representation, I do not see why it is important to emphasize steerability. Moreover, I don’t think steerable G-CNNs are more general per se. Any linear layer is equivariant iff it is a group convolution [bekkers2018, thm1] regardless of the used feature type (one can always convert between regular representations and irreps via a generalized Fourier transform)
>
> **Answer:**  We appreciate the reviewer's perceptive comment. The concept of steerability, as utilized in our paper, is indeed a matter of definition. We employ this term to highlight our ability to select any representation within the hidden layers of the equivariant networks. Importantly, we are not confined to using regular representations beyond the initial hidden layer.
>
> The reviewer's observation that "converting between regular representations and irreps via a generalized Fourier transform" is spot-on. Yet, it is crucial to recognize that the way these networks are parameterized (stemming from the different representations adopted in each layer)  have an impact on the model's implicit bias. For clarity, we refer to Ref. [1], where it's illustrated that even though linear CNNs and linear fully-connected nets can equivalently parameterize all linear predictors, training dynamics of these networks differ significantly, resulting in distinct inductive biases. We will elaborate on this point in the revision to ensure a clear understanding.
>
>
> **Comment 4:** How does the following reference relate to the current work? It is also about generalization bounds of equivariant networks (through PAC learning) and thus seems related. Elesedy, B. Group symmetry in PAC learning.
>
> **Answer:** We thank the reviewer for bringing this paper to our attention. Our understanding is that this work, like many other existing studies that quantify generalization gains for equivariant networks through PAC learning, fundamentally relies on the following reasoning: the invariance of both the target function and the neural networks means that learning is effectively conducted on the quotient space of group orbits. This results in a reduction of the covering number, and hence, the Rademacher complexity of the hypothesis space, which is used to derive the improved generalization bound.
>
> While these approaches can formulate a worst-case generalization upper bound that is universally applicable to all distributions, they may neglect a vital aspect of neural network, i.e., the training process. When this training aspect is taken into consideration, the enhanced upper bounds posited in these works may not necessarily be meaningful. For instance, in our paper, under certain distributions (denoted as $\mathcal{D}\subset \mathbb{R}^{d_0}_G$), learning with or without the equivariant constraint converges to the exact same solution when models are trained using gradient flow. As a result, the imposition of group symmetry does not lead to any improvement in generalization, in contradiction to the findings of the existing works where the training aspect is not accounted for. We believe that our perspective offers a more nuanced understanding that aligns more closely with the behavior of models during the training process.
>
> [1] C. Yun et al. A unifying view on implicit bias in training linear neural networks. ICLR, 2021
>
> [2] K. Lyu and J. Li. Gradient descent maximizes the margin of homogeneous neural networks. ICLR, 2020
>
> [3] Z. Ji and M. Telgarsky. Directional convergence and alignment in deep learning. NeurIPS ,2020.
>
> [4] H. Lawrence et al., Implicit Bias of Linear Equivariant Networks. ICML, 2022.

---

> > ### Comment · Reviewer_nbB9 · 2023-08-21
> >
> > I greatly appreciate the detailed and considerate response. The response as well as those to the other reviewers, has strengthened my belief that this is a solid submission worth of publication at Neurips. I slightly improved my score. Thank you for the great work.
> >
> > Finally, I just want to clarify that it was not my intention to downplay the paper by mentioning the word "trivial", the result seems intuitive which is a good thing (!) and the mathematical rigor is greatly appreciated. I hope elements of the discussion with all the reviews end up in the final paper (I have confidence they will), but even without it, the openreview discussions are a nice compendium to the paper.

---

### Official Review · Reviewer_3ceq · 2023-07-07

**Soundness:** 3 good
**Presentation:** 3 good
**Contribution:** 4 excellent
**Rating:** 6
**Confidence:** 1

**Summary:**

The paper presents a collection of theoretical findings that highlight the equivalence between using a linear steerable network for group-invariant binary classification and employing data augmentation for non-invariant models within the specified context. Additionally, the paper demonstrates that linear steerable networks outperform their non-invariant counterparts in terms of generalization bounds.

**Strengths:**

* The paper builds upon previous studies that have explored the implicit regularization properties of different network architectures. It references works that have examined the convergence of linear predictors trained under gradient descent (GD) towards max-margin support vector machines (SVMs) and the implicit regularization of linear convolutional and tensor networks.

* The paper discusses the design of group-equivariant DNNs that preserve symmetry transformations in both input and output spaces. It highlights the empirical evidence suggesting that group-equivariant steerable networks improve generalization performance, especially in tasks with group symmetry and limited data.

* The paper presents its key result, which demonstrates that linear equivariant steerable networks trained under gradient flow (GF) converge towards the unique group-invariant classifier that maximizes the margin with respect to a norm defined by the input group representation.

**Weaknesses:**

This paper is theoretical in nature. It does not present any new empirical results or experiments to validate its claims. Unfortunately, the reviewer has little knowledge of the topic.

**Questions:**

Please improve the presentation of implications of the research findings and future research directions.

**Limitations:**

The limitations of the theoretical study are adequately discussed.

---

> ### Author Rebuttal · Authors · 2023-08-09
>
> **Comment 1:** This paper is theoretical in nature. It does not present any new empirical results or experiments to validate its claims. Unfortunately, the reviewer has little knowledge of the topic.
>
> **Answer:** We agree with the reviewer that empirical verification of the theory established in this paper could  substantively enhance the impact of our theoretical findings. However, a notable limitation of our findings is that the implicit bias of gradient flow, as studied here, holds true only in an asymptotic sense. The convergence rate to the directional limit is, in fact, exponentially slow. This is consistent with the findings in Ref [1,2]. Specifically, Ref [2] shows that even if the loss function reaches an order of $10^{-5}$, the limit direction of the parameterized classifier (of a linear fully-connected network) is dependent on the initialization. This is because the networks are not trained all the way to zero loss. Understanding the behavior of gradient flow in a non-asymptotic regime is an important direction for future work.
>
> We have mentioned this limitation in the conclusion of our paper, and we will emphasize this point further in the main text of our revised manuscript.
>
> [1] D. Soudry, E. Hoffer, M. S. Nacson, S. Gunasekar, and N. Srebro. The implicit bias of gradient descent on separable data. The Journal of Machine Learning Research, 19(1):2822–2878, 2018.
>
> [2] C. Yun, S. Krishnan, and H. Mobahi. A unifying view on implicit bias in training linear neural networks. In International Conference on Learning Representations, 2021

---

### Official Review · Reviewer_UfQK · 2023-07-08

**Soundness:** 3 good
**Presentation:** 3 good
**Contribution:** 3 good
**Rating:** 7
**Confidence:** 4

**Summary:**

This paper studies the implicit bias on the use gradient flow for the purpose of binary (group-invariant) classification while using linear equivariant steerable networks. There is now a fair amount of work on implicit bias in similar settings in the non-equivariant case. There is also some work for linear equivariant networks. However, as the authors note, it is unclear whether a steerable network, trained by gradient descent, can achieve a minimizer with some parameter norm which is comparable to non-equivariant cases. The main results essentially show that linear steerable networks trained on some data converge to the same direction as a fully connected NN trained on augmented data. Moreover, such a network trained on the original data always attains a wider margin than a fully connected network trained on augmented data. An additional result is shown that a linear steerable network trained using gradient flow has better generalization than its non-equivariant counterpart when the data has an group invariant distribution. Section 3 describes the background and results formalizing and proving some of the above points are shown in later sections.

Minor comments:
- On page 3, line 122: "Equivalently, we want the following diagram to be commutative.." --> reword this since that diagram is already commutative, but that you want your network to adhere to it.

**Strengths:**

- The results, to the best of my knowledge, are solid and represent a good contribution complementing a host of body of work for non-equivariant models.
- Generally speaking the paper is well written (although there is a flip side to this, which I will write in the weakness) and has a non-trivial contribution.

**Weaknesses:**

- One issue with the paper is its density. A lot of time is spent on background that doesn't need to be in the main paper. I understand the authors want to be thorough and that this is difficult. However, the essential idea and the proof techniques that the authors use, and the literature that it builds upon, are well-known, hence so much background is not necessary. I would suggest putting some of this material in the appendix and expanding on the results presented in the paper better and also comparing them to existing results.
- Related to the above. I think the writing can be made more easy going (even if I had no problem in following it). I would understand for certain problems such heavy formalization is necessary. But I think this particular problem doesn't need it. It could be written simply. But in any case I would like to see more discussion about the actual results than the background. If the authors think so much of the background is necessary, I would be curious to hear their arguments. I think we can drop describing what a group or representation is, for example. It doesn't have to be done in every paper.

**Questions:**

- Please improve the discussion around the work of Lawrence (2021) at the end of page 2 and beginning of page 3. I am unable to parse the comments here. How are they "not actually invariant or equivariant" just because "input is on a group"? The inputs can be on a group in general in equivariant networks, or on spaces on which a group acts. I am also confused with the random claim that the paper works with Euclidean inputs with non-transitive actions. I don't think the formalism here is more general than operating on homogeneous spaces. Do the authors mean something else? I would appreciate a clarification.

**Limitations:**

Yes

---

> ### Author Rebuttal · Authors · 2023-08-09
>
> **Minor Comment**: On page 3, line 122: "Equivalently, we want the following diagram to be commutative.." --> reword this since that diagram is already commutative, but that you want your network to adhere to it.
>
> **Answer**: We thank the reviewer for pointing this out. Yes, the diagram has indicated commutativity and we will reword it in the revised manuscript.
>
> **Comment 1 and 2**: One issue with the paper is its density. A lot of time is spent on background that doesn't need to be in the main paper.  I would suggest putting some of this material in the appendix and expanding on the results presented in the paper better and also comparing them to existing results.
>
> **Answer**: We thank the reviewer for the thoughtful suggestion. In line with this guidance, we will relocate some of the relevant but not strictly essential group-theoretical background information to the appendix. This will allow us to streamline the main text. In its place, we will improve the readability of the paper by including additional remarks and discussion, aimed at comparing our results with existing works in the field.
>
> **Comment 3**: Please improve the discussion around the work of Lawrence (2021) at the end of page 2 and beginning of page 3. I am unable to parse the comments here. How are they "not actually invariant or equivariant" just because "input is on a group"? The inputs can be on a group in general in equivariant networks, or on spaces on which a group acts. I am also confused with the random claim that the paper works with Euclidean inputs with non-transitive actions.
>
> **Answer**: In the work of Lawrence et al. (2021), the inputs to their networks are functions over a group $G$, and the input group action is characterized by the regular representation. Specifically, the input space is defined as $X_0 =$ {$f:G\to\mathbb{R}$}, with the input action given by
> $$(\rho_0(g))f(g') = f(g^{-1}g'), \forall g\in G, \forall f\in X_0$$
> Upon inspection, one can prove that, under their setting, $G$-invariant linear functions  $\Phi:X_0\to\mathbb{R}$ are constrained to the (trivial) form:
> \begin{equation*}
>     \Phi(f) = C \sum_{g\in G}f(g),
> \end{equation*}
> where $C$ is a multiplicative constant.
>
> To avoid learning this trivial function, Lawrence et al. (2021) chose to parameterize $\Phi$ using a group-convolutional neural network, in which the final layer is replaced by a fully-connected (FC) layer. While this FC layer provides the capability to learn more complex and nontrivial functions, it simultaneously undermines the property of group invariance.
>
> On the other hand, in our paper, the input space is the Euclidean space $X_0 = \mathbb{R}^{d_0}$, and there exist non-trivial linear maps that are invariant to the input group action. We will include an explicit explanation in the revised version of our paper.

---

> > ### Comment · Reviewer_UfQK · 2023-08-10
> > **Thanks**
> >
> > Thanks for the clarification and rebuttal. I hope that a few of these (seemingly minor) points will be explicitly addressed. Since a few things I had questions about have been clarified, I will raise my score accordingly.

---

> > > ### Comment · Reviewer_UfQK · 2023-08-10
> > > **Clarification**
> > >
> > > Just to clarify to the authors, in order to not provoke any kind of anxiety. I will raise my score when I am able. The system currently does not allow me to do so. I also found the clarification w.r.t Lawrence et al. useful because now I am able to see that the work certainly goes quite further and is more complete, and is thus a good contribution.

---

> > > > ### Author Response · Authors · 2023-08-15
> > > > **Thanks**
> > > >
> > > > We thank the reviewer for reading through our response and kindly raising the score.

---

### Official Review · Reviewer_vUEN · 2023-07-10

**Soundness:** 2 fair
**Presentation:** 3 good
**Contribution:** 3 good
**Rating:** 5
**Confidence:** 3

**Summary:**

The paper considers the implicit bias of gradient flow optimization for linear equivariant steerable networks. The paper consists of the following steps and contributions:
* The equivariant network is assumed to have regular representation at the input (Assumption 3.1). The architecture is characterized in Proposition 3.2, and is based on an overparameterized linear function applied to group averaged input.
* The space of G-invariant linear classifiers is characterized in Proposition 3.6, which can be seen as the range of the adjoint of group projection operator. (same as the group projection for unitary representations)
* The analysis of gradient descent is done based on using gradient flow (eq 25). Assuming the existence of G-invariant separating function (Assumption 3.7), Theorem 4.1 shows that the gradient flow converges to a maximum margin G-invariant classifier.
* A fully connected linear network with data augmentation converges to the solution \textbf{direction} for linear steerable network for unitary input representation (Corollary 5.1), and with modified loss for non-unitary input representation (Corollary 5.4).
* Despite sharing the direction, the margin is still larger for the steerable model (Theorem 6.1).
* Finally, the generalization bound for steerable and non-steerable models are obtained using classical results for max-margin SVM from the book of S. Shalev-Shwartz and S. Ben-David. (Theorem 6.3)

As I mention below, the paper has a collection of interesting result in a previously unexplored topic. However, I have some technical concerns about the definition of the model, and would like to hear the authors’ answers.


**Strengths:**

The paper considers the problem of implicit bias for equivariant networks using gradient flow analysis, which was not previously studied in the literature. Although the results seem to be mostly based on s. techniques of Lyu and Li [2020] and Ji and Telgarsky [2020], the conclusions are interesting, in particular, the equivalence of data augmentation and model equivariance in terms of converged direction.

**Weaknesses:**


* I have some concerns about the choice of linear model for equivariant networks. When considering linear group invariant classifiers, the steerable network, based on Schur’s lemma, utilizes only the trivial representation (See M. Weiler and G. Cesa 2019). The authors noticed a similar issue with Lawrence et al. 2021 paper, as they mentioned “In their case, the only group-invariant linear classifier is the group average of the input up to a scalar multiplication, due to the transitivity of the regular representation”. To summarize, a linear invariant network with equivariant kernels works only with trivial representations. In other words, only a single basis in equation (4), the one corresponding to the trivial irrep, matters for the optimization and the output. It seems to me that the current definition in (4) can be misleading in this sense. Note that Proposition 3.2 indicates as well that the network is only dependent on a group averaged version of $\mathbf{x}$.
* It seems that the equation (7) does not define an equivariant map. If we assume a regular representation,  then the map is equivariant only if $\mathbf{w}_1$ has a group circulant structure. With this additional required structure, Lemma B.1 in the supplementary material cannot be directly applied, because of entry repetition in $\mathbf{w}_1$. Note that according to assumption 3.1, the regular representation and $G$-lifting map is assumed for the first later.
* The independence of the generalization error from group size in Remark 6.5 should be considered with care. Note that there is an experimental evidence, although light, for the impact of group size on the generalization (see Sokolic et al). On the other hand, the group size can shrink the size of the space $\mathbb{R}^{d_0}_{G}$, and can potentially reduce $\overline{R}$ proportional to the group size. I feel that Remark 6.5 should be a bit more nuanced.


**Questions:**


* Does not the choice of regular representation for the input impose that $d_0$ is a function of group size? The steerable network of Cohen and Welling 2017 is more general and includes translation operations and features fields.
* How does the result generalize to compact but not finite groups? For example, the data augmentation for compact inifinite groups seems to be non-trivial. Some discussions on this point can be useful for the paper.
* There are some follow-up works on the generalization bounds for equivariant networks with implicit and explicit bounds on the error. It is good to discuss the connection and compare the bound. It seems that all these works relate the generalization gain to the group size, which is different  from the current paper.
   * Sicheng Zhu, Bang An, and Furong Huang. Understanding the Generalization Benefit of Model Invariance from a Data Perspective. In Advances in Neural Information Processing Systems, 2021
  * Clare Lyle, Mark van der Wilk, Marta Kwiatkowska, Yarin Gal, and Benjamin Bloem-Reddy. On the benefits of invariance in neural networks. arXiv preprint arXiv:2005.00178, 2020
  * Arash Behboodi, Gabriele Cesa, Taco Cohen, A PAC-Bayesian Generalization Bound for Equivariant Networks, In Advances in Neural Information Processing Systems, 2022




**Limitations:**

See above comments.

---

> ### Author Rebuttal · Authors · 2023-08-09
>
> **Comment 1**: When considering linear group invariant classifiers, the steerable network, based on Schur’s lemma, uses only the trivial representation. I.e., only a single basis in equation (4) matters for optimization and the output. The definition in (4) is misleading in this sense. Proposition 3.2 indicates that the network is only dependent on a group averaged version of x.
>
> **Answer**: We thank the reviewer for the insightful comment. You are correct: based on Schur's lemma, the final linear predictor parameterized by a multi-layer linear steerable network uses only the trivial representation, due to the trivial representation on the output layer.
>
> However, we emphasize that although the parameterized linear predictor has simple form, the manner in which this predictor is parameterized has huge impact on the optimization, and consequently, the implicit bias. E.g., in Ref. [1]: even though linear CNNs and linear fully-connected nets equivalently parameterize all linear predictors, they have distinct inductive biases.
> We also emphasize that although the output representation is the trivial representation, we do not limit the selection of representations for hidden layers. Hence multiple bases can coexist in equation (4).
>
> [1] C. Yun et al.. A unifying view on implicit bias in training linear neural networks. ICLR, 2021
>
> **Comment 2**: Equation (7) is not equivariant.
>
> **Answer**: We provide a simple proof: Eq.(7) reads
> $$\Psi_1^{\text{G-CNN}}(x, w_1)(g) = w_1^\top g^{-1}x.$$
> We need to check:
> $$ \Psi_1^{\text{G-CNN}}(hx, w_1) = \rho_1(h) \Psi_1^{\text{G-CNN}}(x, w_1), \forall h\in G$$
> The LHS when evaluated at $g$, is:
> $$ \Psi_1^{\text{G-CNN}}(hx, w_1)(g) = w_1^\top g^{-1}hx.$$
> The RHS is
> $$ \Psi_1^{\text{G-CNN}}(x, w_1)(h^{-1}g) = w_1^\top (h^{-1}g)^{-1}x.$$
> Hence it is equivariant.
>
> **Comment 3**: The independence of the generalization error from group size in should be considered with care. Experimental evidence exists for the impact of group size on generalization (Sokolic et al). The group size can shrink $\mathbb{R}_G^{d_0}$, and can potentially reduce $\bar{R}$.
>
> **Answer**: The reviewer is correct that the situation is more nuanced. In Theorem 6.3, we show that the improvement in generalization is **directly** tied to the reduction from $R$ to $\bar{R}$, on which the group size may or may not have an impact. E.g., consider the input space $\mathbb{R}^3$, on which $G=SO(2)$ acts by rotating around the z-axis. Regardless of the cyclic subgroup $C_n$ of $G$, as long as $n\ge 2$, the averaged input $\bar{x}$ is always the same for any $x$, i.e., $\bar{x}$ is the projection of $x$ onto the z-axis. Hence $\bar{R}$ is the same, no matter the group size $|C_n|=n$.  However, that there are certainly cases where the group size will (indirectly) affect generalization by reducing $\bar{R}$.
>
> We emphasize that this conclusion is reached **only** after considering the role of training in NNs. This understanding is largely absent in most existing research that attempts to quantify the generalization gain when learning under symmetry. Regarding the comparison with Sokolic et al., we will provide a detailed response to another question below.
>
> **Comment 4**: Does not the choice of regular representation for the input impose that $d_0$ is a function of group size?
>
> **Answer**: The regular representation is chosen only for the first hidden layer, while the input $G$-representation is not restricted.
>
> **Comment 5**: How does the result generalize to compact but not finite groups? For example, the data augmentation for compact inifinite groups seems to be non-trivial.
>
> **Answer**: Our current work contains a restriction that hinders us from directly extending the result to compact groups. This limitation is Assumption 3.1, which mandates that the first layer is a
> $G$-lifting map, with the first-layer features explicitly represented as matrices. Should this constraint be removed, we would be poised to expand our result to encompass compact groups.
>
> **Comment 6**: There are some follow-up works on the generalization bounds for equivariant networks. It is good to discuss the connection and compare the bound. It seems that all these works relate the generalization gain to the group size, which is different from the current paper.
>
> **Answer**: Many existing works that relate generalization gains for equivariant networks to group size are founded on this reasoning: the invariance of the target function suggests that learning operates on the quotient space of group orbits. This leads to a decrease in the covering number, and consequently, the Rademacher complexity. This reduction is connected to the group size. For instance, in Sokolic et al., the data distribution is assumed to be confined to $X = G\times X_0$, so $G$-invariant models are essentially learning on $X_0$, resulting in a $|G|$-fold decrease in covering numbers.
>
> While these methods can establish a worst-case generalization upper bound that applies universally across distributions, they overlook a critical aspect, i.e., the training of neural networks. Notably, under specific distributions (those with $\mathcal{D}\subset \mathbb{R}^{d_0}_G$ in our paper), learning with or without an equivariant constraint when trained under GF converges to the same solution. Consequently, imposing group symmetry does not yield any generalization gains.
>
> A related example involves training w/ and w/o data augmentation. If a data distribution is such that all samples remain unchanged after augmentation, then the use of data augmentation provides no advantage. Our generalization result, as articulated in Theorem 6.3, clarifies that it is not merely the group size but the distance between the distribution $\mathcal{D}$ and the subspace $\mathbb{R}^{d_0}_G$ hat influences the enhanced generalization of equivariant networks. This insight emerges when the training aspect is factored into the analysis.

---

### Official Review · Reviewer_WwNm · 2023-07-23

**Soundness:** 3 good
**Presentation:** 3 good
**Contribution:** 3 good
**Rating:** 6
**Confidence:** 2

**Summary:**

This paper analyzes the implicit bias of gradient flow on linear group-equivariant steerable networks for group-invariant binary classification. It shows that the parameterized predictor converges in a direction that aligns with the unique group-invariant classifier with a maximum margin that is defined by the input representation. Under a unitary assumption, this paper also suggests the equivalence between data augmentation used in non-invariant models and learning with steerable networks. When the underlying distribution is group-invariant, linear steerable networks outperform their non-invariant counterparts in terms of improved margin and generalization bound.

**Strengths:**

1. The paper is organized in a way easy to read and follow. The authors provide sufficient background knowledge for readers to understand Equivariant Steerable Networks. And all the assumptions are stated clearly in the problem setup section.
2. The contributions of the paper have been listed in a rather detailed and clear way in the paper. The authors point out their main findings in theorems, followed by explanations in remarks and close each section with corollaries.
3. The findings presented in this paper explain the success of equivariant steerable networks to a certain extent.

**Weaknesses:**

1. The authors claim to fully characterize the implicit bias of the training algorithm. However, there is no experiment on empirical dataset other than theoretical results throughout the paper. It would be more promising if the conclusion can be supported by empirical results.

**Questions:**

Are these findings limited to binary classification?

---

> ### Author Rebuttal · Authors · 2023-08-09
>
> **Comment 1**: The authors claim to fully characterize the implicit bias of the training algorithm. However, there is no experiment on empirical dataset other than theoretical results throughout the paper. It would be more promising if the conclusion can be supported by empirical results.
>
> **Answer**: We totally agree with the reviewer that empirical verification of the theory established in this paper could  substantively enhance the impact of our theoretical findings. However, a notable limitation of our findings is that the implicit bias of gradient flow, as studied here, holds true only in an asymptotic sense. The convergence rate to the directional limit is, in fact, exponentially slow. This is consistent with the findings in Ref [1,2]. Specifically, Ref [2] shows that even if the loss function reaches an order of $10^{-5}$, the limit direction of the parameterized classifier (of a linear fully-connected network) is dependent on the initialization. This is because the networks are not trained all the way to zero loss. Understanding the behavior of gradient flow in a non-asymptotic regime is an important direction for future work.
>
> We have mentioned this limitation in the conclusion of our paper, and we will emphasize this point further in the main text of our revised manuscript.
>
> [1] D. Soudry, E. Hoffer, M. S. Nacson, S. Gunasekar, and N. Srebro. The implicit bias of gradient descent on separable data. The Journal of Machine Learning Research, 19(1):2822–2878, 2018.
>
> [2] C. Yun, S. Krishnan, and H. Mobahi. A unifying view on implicit bias in training linear neural networks. In International Conference on Learning Representations, 2021

---

> > ### Comment · Reviewer_WwNm · 2023-08-16
> > **Thanks!**
> >
> > thanks for the explanation, although I think some kind of experiments will help.

---

### Decision · Program_Chairs · 2023-09-21

**Decision:**

Accept (poster)

**Comment:**

The authors are studying the implicit bias of gradient flow on linear group-equivariant steerable networks for group-invariant binary classification. The results regarding the equivalence with data augmentation have been well-received. All the reviewers have found the work to be relevant and interesting, with meaningful theoretical results. The reviewers have highlighted several aspects for improving the paper's presentation. We suggest that the authors incorporate these suggestions into the final camera-ready version.
In conclusion, I recommend accepting this paper.